# A Review of Research Progress on the Performance of Intelligent Polymer Gel

**DOI:** 10.3390/molecules28104246

**Published:** 2023-05-22

**Authors:** Shuangchun Yang, Zhenye Liu, Yi Pan, Jian Guan, Peng Yang, Muratbekova Asel

**Affiliations:** 1Department of Petroleum and Natural Gas Engineering College, Liaoning Petrochemical University, No. 1, West Section of Dandong Road, Wanghua District, Fushun 113001, China; yangchun_bj@126.com (S.Y.); wsymhgdyjs@163.com (Z.L.); 2Engineering Department of Greatwall Well Drilling Company, China National Petroleum Corporation, Panjin 124000, China; gj_04081@163.com (J.G.); shiyan_1997@163.com (P.Y.); 3Institute of International Education, Liaoning Petrochemical University, Fushun 113001, China; asel_1209@bk.ru

**Keywords:** intelligent polymer gel, responsiveness of the gels, environment of external stimulation, different configurations, research progress

## Abstract

Intelligent polymer gel, as a popular polymer material, has been attracting much attention for its application. An intelligent polymer gel will make corresponding changes to adapt to the environment after receiving stimuli; therefore, an intelligent polymer gel can play its role in many fields. With the research on intelligent polymer gels, there is great potential for applications in the fields of drug engineering, molecular devices, and biomedicine in particular. The strength and responsiveness of the gels can be improved under different configurations in different technologies to meet the needs in these fields. There is no discussion on the application of intelligent polymer gels in these fields; therefore, this paper reviews the research progress of intelligent polymer gel, describes the important research of some intelligent polymer gel, summarizes the research progress and current situation of intelligent polymer gel in the environment of external stimulation, and discusses the performance and future development direction of intelligent polymer gel.

## 1. Introduction

At present, the intelligent polymer gel is a thermal polymer material, which can respond to environmental stimuli by changing its structure to adapt to changes in the external environment.

In recent years, more and more attention has been paid to responsive intelligent polymer gels. Due to the increase in functional monomers and monomer macromolecules, the field of development has expanded. Compared to the traditional gels, responsive intelligent polymer gels can feel external environmental stimuli, such as temperature, light, electric field, pH, chemical substances (glucose), mechanical stress [1,2], magnetic field [3,4], and salt [5,6], and can adapt to environmental changes by changing their structure. Different intelligent polymer gels have different application fields: traditional intelligent polymer gels are mainly based on synthetic polymers, the synthetic process of this kind of material is complex and the cost is high; the important factors that restrict the development of such gels are poor biocompatibility and not being easy to degrade; therefore, the application development field is limited [7,8,9,10,11]. However, a variety of intelligent polymer gels made from natural polymer materials can solve these shortcomings perfectly; with good biocompatibility, easy degradation, and other excellent performance, they can play a more critical role in drug controlled release, tissue engineering, biomimetic intelligent materials, and other fields [12,13]. The gels are made of synthetic polymer materials and are natural to their structure. Based on the intelligent features of intelligent polymer gel, in recent years, the demand for polymer gels in various fields has been extensively studied. Intelligent polymer gels with related properties were prepared according to the requirements, giving them good prospects in biological engineer materials and focused directions, and while their advantages have grown, their defects have also expanded. In bioengineering, serious problems are often encountered (e.g., transparency issues, the inevitable risk of disease transmission from the donor to the recipient, and undesired inflammation and immune reactions), and clinical outcomes of amniotic membrane pattern grafts may be limited. Therefore, cytocompatible hydrogels may alternatively provide a safer approach to treat ocular surface disease. This approach could be further used for safe engineering of other ocular cells/tissues within the posterior segment (e.g., to treat retinal diseases), so it is a necessity to study hydrogels in ocular in vitro cellular models and in vivo biocompatible animal models. In recent years, with the deepening in research, more and more researchers have gradually solved these defects, and all kinds of intelligent polymer gels with good mechanical properties, good biocompatibility, and easy degradation have been successfully prepared. Kollarigowda et al. [14] from Canada prepared NIPAM-HA-MA (N-Isopropylacrylamide, hyaluronic acid (HA), Methacrylic anhydride (MA)) hydrogel based on NIPAM thermosensitive hydrogel. Its mechanical properties decreased when the temperature increased from 25 °C to 35 °C, and its rigidity increased when the temperature was lower (as shown in Figure 1).

It is biocompatible with human dermal fibroblast cell line (HDF), and the cell adhesion density reaches its maximum at 37 °C, but decreases at 25 °C. It can be used as a biomaterial to simulate cell induction in tissue engineering applications. Samiullah Khan et al. [15] from Pakistan prepared an injectable pH/thermal dual-reaction injectable hydrogel with cytocompatibility, and it could be crosslinked in situ at pH 2.1 and temperature 25 °C, which showed the highest release and good cytocompatibility; the injectable formulations developed can serve as a control reservoir, administered subcutaneously, for long-term treatment of other diseases. Mizuguchi et al. [16] from Japan developed temperature-responsive multifunctional hydrogels from genetically engineered proteins, which have a controllable sol-gel transformation, superior transparency, adjustable mechanical, functional biological characteristics, and growth factor transfer activity. These properties will contribute to designing three-dimensional microenvironments for Spatio-temporal control of specific cell functions. Tian Li et al. [17] synthesized a series of ethylene glycol-based triblock copolymers; they fit nicely into cells and have soft, rubbery properties. The maximum concentration is 20 wt%, and the temperature is 37 °C. Within 5 s, the mechanical gel formed is strong, and it can be used as an injectable scaffold for cell culture and tissue engineering applications. To obtain hydrogels with high electric field response performance, Tu Xin et al. [18] selected environment-friendly natural polymer materials (starch and gelatin) as raw materials to prepare GO/natural polymer composite hydrogels by adding graphite oxide (GO) into the system.

With starch hydrogels as the matrix, a series of GO/starch hydrogels were prepared (as shown in Figure 2), and gelatin/GO hydrogels with different GO contents were designed and synthesized (as shown in Figure 3). The addition of GO improves the mechanical strength and toughness of the two gels, which can be better applied to the field of bioengineering.

Smart polymers (SP) have become an important class of polymers and their applications are increasing in various fields. SP, also known as irritant soluble–insoluble polymers or environmentally sensitive polymers, have been widely used in the biotechnology, medical, and engineering fields [19].

Smart polymers or stimulus-responsive polymers undergo reversible, large physiological or chemical changes in their properties due to small environmental changes [20]. They can respond to single or multiple stimuli corresponding to temperature, pH, electric or magnetic fields, light intensity, biomolecules, etc., inducing macroscopic responses in the material such as swelling, collapse, or transition from solution to gel, depending on the physical state of the chain [21]. Linear and solubilized smart macromolecules move from single phase to biphasic near transition conditions, producing a reversible sol-gel state. Smart cross-linked networks undergo chain reorganization under transition conditions and the network changes from a collapsed to a swollen state. Smart surface stimulation alters its hydrophilicity to provide a responsive interface. All these variations can be used to design smart devices for multiple applications, for example, minimally invasive injection systems [22], pulsatile drug delivery systems [23,24], or new devices for cell culture or tissue engineering matrices [25].

Smart polymers can be immobilized or grafted on solid surfaces by cross-linking to form hydrogels, or dissolved in water. When a smart polymer is subjected to a series of stimuli (e.g., by increasing the temperature above a certain threshold), the polymer chains change from water-soluble to water-insoluble, thus converting the polymer material from hydrophilic to water-insoluble. The polymeric material is converted from a hydrophilic to a hydrophobic state [26,27]. Depending on the polymer system, the reaction may be precipitation, gelation, adsorption, collapse of the polymer attached to the surface, or hydrogel collapse, and the driving force behind these reversible transformations varies with the stimulus [28].

A hydrogel is a three-dimensional (3D) network of natural or synthetic polymers. It can absorb large amounts of water, up to several thousand times its dry weight. Hydrogels are three-dimensional (3D) natural or synthetic polymer networks that can absorb large amounts of water in water without dissolving the hydrophilic but cross-linked structure due to its hydrophilic but cross-linked structure [29,30]. These these hydrogels can be chemical hydrogels, which are covalently linked polymeric meshworks, or physical hydrogels, in which the formation of the network is reversible. Interestingly, “smart” hydrogels are defined as hydrogels that undergo a sudden reversible volumetric phase transition (VPT) or sol-gel phase transition in response to small external changes (stimuli) in environmental conditions [31]. This response is relatively fast compared to conventional hydrogels. Smart hydrogels constitute a new generation of hydrogels that are currently being developed for a large number of possible applications, including smart actuators for chemical valves, biosensors, optical systems, templates for nanoscale devices, scaffolds for tissue engineering, artificial muscles, carriers with drug delivery, soft bionic machines, and substrates for bioseparation [32,33,34,35,36,37]. However, important challenges remain with smart hydrogels, such as slow re-response times to stimuli and the hysteresis phenomena associated with them. Synthetic hydrogels are subject to degradation upon exposure to appropriate environmental stimuli. When it is exposed/implanted in the human body, synthetic hydrogels will degrade due to appropriate environmental stimuli, as well as the preparation of hydrogels that mimic the extracellular matrix (ECM). Preparation of hydrogels that mimic ECM to make them more biological compatible to prevent response of the immune system [38].

Among the recent advances in biomedical devices, one of the major limitations of hydrogels in stimulus responsive applications is the limited diffusion rate. This can be avoided by designing interconnected pores in the polymer structure to avoid the formation of capillary networks in the polymer structure, and by reducing the size of the hydrogel to greatly reduce the diffusion path. Reducing the lag time for inducing smart reactions is useful in biomedical devices such as sensors and actuators [38].

The reactivity of polymers or gels is usually obtained by cross-linking specific smart molecules to their chain networks [39], or by inducing directional or positional organization at the molecular level [40]. Physical, chemical, or directional changes triggered by appropriate environmental stimuli are transferred into the molecular meshwork. Physical, chemical, or directional changes triggered by environmental stimuli are transferred into the network and therefore detected as physical or chemical changes at the contiguous level. It is worth mentioning the development of the so-called mechanical hoof compounds, which are capable of responding to mechanical stresses by changing the color of the emitted visible light at visible wavelengths when inserted into the polymer network [41]. This reaction arises from changes in atomic binding corresponding to the different chemical states of the compound; an example of this property is the colorless spirulina molecule, which becomes colored in the plumbagin state by a reversible reaction in response to stimuli such as light, temperature, metal ions or mechanical stress [42]. However, another class of reactive molecules capable of changing their geometric conformation has been synthesized; changes in the shape and size of the molecule are due to changes in the position of the constituent atoms without altering their bonds, so that their chemical properties are maintained [43,44]. When properly bonded to a polymer network, their collective changes are reflected as mesoscopic reactions of the material. The mesoscopic response of the material appears through some detectable mechanical changes, such as volume expansion or contraction [45].

With the refined functional requirements of various fields, the research of intelligent polymer gels has gradually developed from single response type to multi-response type, and many gels with more complete functions and better properties have been prepared and developed. Currently, some magnetic gels can achieve significant contraction. Their unique magnetoelastic properties give them the potential to mimic muscle contraction [46]. The magnetic field-sensitive smart polymer gel is composed of a three-dimensional network of polymers and a magnetic fluid. Because of the magnetic properties of the magnetic fluid and the interaction between the fluid and the polymer chains, the polymer gel expands and contracts in response to the magnetic field. When the magnetic fluid is immobilized as a cohesive, the intrinsic properties of the magnetic fluid are difficult to see. Similar to the case of magnetic powder immobilization, when an aqueous-based magnetic fluid is encapsulated in a polymer gel, it still retains its extraordinary magnetic properties and exhibits a stretching behavior in the magnetic field direction. It is possible to obtain smart polymer gels that are very sensitive to magnetic stimulation by adjusting the magnetic fluid content, crosslinking density, and other factors [47].

In this paper, the preparation methods, response properties, and application fields of intelligent polymer gels were reviewed from two aspects of stimulation response types and environmental factors.

## 2. Single Response Intelligent Gel

Single response intelligent polymer gel only responds to a specific external environmental stimulus: depending on the type of external stimulus, it can be divided into physical stimulus-response type and chemical stimulus-response type [47].

### 2.1. Physical Stimulus Responsive Gel

#### 2.1.1. Temperature Response Type

Due to their simple stimulation form, high response efficiency, and excellent mechanical strength, temperature-responsive intelligent gels have been widely used in flexible actuators, artificial muscles, transport, and release of loads, intelligent surfaces, biological organs, and other essential fields [48,49]. However, the current situation of temperature-responsive intelligent gels brings higher requirements for them: it must take into account the comprehensive performance of high load capacity, diversified response modes, high response sensitivity, precise structure form, and diversified and controllable self-driven deformation, which are also the technical problem to be solved urgently.

Hydrophobic groups (such as methyl, ethyl, and propyl groups) and hydrophilic groups (such as amides and carboxyl groups) are present in the monomer. Hydrophilic groups are combined with water through hydrogen bonds at low temperatures, which causes the hydrogel to expand and present a solution state. When the temperature rises close to the human body temperature, hydrogen bonds will decrease, and hydrogel will shrink and show a gel state.

Tanaka et al. [50] found that the volume transformation temperature of P-N-isopropylacrylamide (PNIPAM) was about 32 °C. Destroying the balance between hydrophilic and hydrophobic groups in the macromolecular chain of PNIPAM, causes the polymer to release water from the hydrophobic interface, resulting in polymer precipitation [51]. It is easy to control, easy to modify, and has other excellent properties, and it has become the most popular type of temperature-sensitive gel studied at present.

To effectively improve the response speed, mechanical properties, low critical dissolution temperature of PNIPAM gels, and better meet the biomedical applications, such as the controlled release of drugs, separation, and immobilization of enzymes. Among the smart polymers, thermoresponsive polymer PNIPAM is very significant because of its well-defined structure and property, especially its temperature response which is close to human body and can be finetuned. Mechanical properties are critical for the performance of stimuli responsive hydrogels in diverse applications. So far most of the attempts to achieve this goal have relied on single mechanism, e.g., either interpenetrating network, slide ring or multi-functional crosslinkers. Combination of these methods will direct the future research on PNIPAM based tough hydrogel [52].

A series of biodegradable poly(ethylene glycol)-poly(-caprolactone)-poly(ethylene glycol) (PEG-PCL-PEG, PECE) copolymers triblock copolymers were successfully synthesized and characterized by Fourier transform infrared spectroscopy (FT-IR), nuclear magnetic resonance analysis (^1^H NMR), gel permeation chromatography (GPC), and differential scanning calorimetry (DSC). The PECE copolymers aqueous solution undergoes sol–gel–sol transition as the temperature increases. In addition, the sol–gel–sol phase transition behavior of the copolymers aqueous solutions was deter-mined using the test tube inverting method. The sol–gel–sol transition behavior of the copolymers depended on a number of factors, such as hydrophilic/hydrophobic balance (PEG/PCL ratio) in the molecular structure, topology of the triblock copolymers, and the solution composition of the hydrogel. As a result, the temperature range of phase transition could be varied, which might be useful for its application in many fields, such as drug delivery [53].

It is not easy to find gels that meet these conditions of high mechanical strength, high molding efficiency, and high preparation accuracy at the same time, and most gels can only satisfy one of them. Therefore, a model can be designed by the characteristics of its microstructure to reveal the corresponding deformation mechanism, which provides a new solution to solve the bottleneck problem of thermosensitive smart hydrogels with high mechanical strength, multi-morphological smart response and high precision molding preparation at the same time. Ning Luping [54] used N-isopropylacrylamide (NIPAM) as a monomer, synthetic lithium magnesium silicate (XLG) as cross-linking agent, and nano-wood pulp cellulose as the reinforcing phase, and prepared a thermosensitive bionic intelligent hydrogel with mechanical strength, high molding efficiency, high preparation accuracy, and high intelligent response characteristics through the “one-step” mold molding technology and 3D printing technology. The study found that: Nfc10-base-uv35 with cellulose content of 10 mg/mL in nanomaterials was the best material system for preparing thermosensitive intelligent hydrogels by 3D printing. The bionic thermosensitive intelligent hydrogels were prepared by 3D printing with photosensitive initiator content of 35 mg/mL and irradiation time of 3 min; it has reversible self-driving spiral deformation characteristics under the temperature driving conditions, which effectively realizes the self-driving functional characteristics of the biological template and realizes the efficient reproduction of bionic functions.

To study the factors that affect the ability to load and release drugs, Zhang Zhaofu et al. [55] polymerized NIPAM with crosslinked agent Mesoporous silica nanoparticles (MSN) and initiator Potassium persulfate (KPS), and prepared PNIPA/MSN composite hydrogel with a rich honeycomb structure. MSN, as a cross-linking agent, produces double radicals after its exposure, which interact with the polyhydrocarbon resin phase and form bridge bonds between polymer molecular chains, changing them into insoluble substances with three-dimensional structures. The results showed that the Lower Critical Solution Temperature (LCST) of hydrogel samples was about 33 °C; the temperature sensitivity of compound hydrogels to BSA (bovine serum albumin), egg white lysozyme, and collagen oligopeptides with large, medium, and small molecular weights was similar, and the drug release could be controlled by temperature regulation; The hemolysis test showed that its biocompatibility was excellent. Therefore, PNIPA/MSN composite hydrogel can be used as a temperature-sensitive drug-controlled release carrier.

High-temperature-sensitive hydrogels have good development potential in the low-temperature ranges. Li Tang et al. [56] designed, synthesized and characterized tricopolymer poly (N-tert-buty l acrylamide-co-N-isopropyl acrylamide-co-acrylamide) P (NTBAM-co-NIPAM-co-AM) hydrogel. Due to its high-temperature sensitivity (2 °C), large volume desorption (90%), and low-temperature response range (8 to 10 °C), a novel hydrogel sphere actuator was demonstrated as a temperature indicator to monitor small temperature-induced dimensional changes (as shown in Figure 4).

This design strategy is generally applicable to other temperature-sensitive hydrogels with different shapes and temperature response ranges, and it can be used as cryogenic indicators in a variety of applications, including vaccine storage and transportation.

Temperature-responsive polymer gels can produce volume phase changes by stimulating external temperature changes. Such temperature stimulation changes not only exist naturally, but also can be easily controlled artificially, the gel temperature of some temperature-sensitive gels is close to the human body, which can be applied in bioengineering, drug-controlled release system, and other fields, and that is one of the hot exploration directions in recent years [57,58].

#### 2.1.2. Light Response Type

Photoresponsive hydrogels are composed of polymer networks and photosensitive groups as a functional part, and their physical and chemical properties can be changed by light stimulation. The optical signal is first captured by a photochromic molecule that converts the light irradiation signal into a chemical signal through a photoreaction of isomerization (cis, on-off), cleavage, and dimerization, whose signal is transferred to the functional part of the hydrogel and controls its properties.

The light-responsive gel can undergo physical or chemical changes because of light stimulation. Commonly used photosensitive compounds include chlorophyllic acid, dichromates, aromatic azides and diazo compounds, aromatic nitro compounds and organohalogen compounds. At present, the response mechanism of light-responsive hydrogel has the following three types: add light-sensitive compounds that can be decomposed by light into polymer gel, under the stimulation of light, a large number of ions are generated inside the gel, causing a sudden change in the osmotic pressure inside the gel, and the solvent diffuses from outside to inside, prompting the gel to undergo a volumetric phase transition and producing a photosensitive effect; add light-sensitive compounds into temperature-sensitive gel, and when the gel absorbs certain energy of photons, the light-sensitive compounds convert light energy into heat energy, causing the internal temperature of the gel to increase. When the gel absorbs p hotons of certain energy, the photosensitive compound converts the light energy into heat energy, which increases the internal temperature of the gel, and when the temperature rises to the phase transition temperature of the gel, the gel will swell or shrink and undergo a volumetric phase transition; a more common method is to introduce photosensitive groups into the main or side chains of the polymer. These photoreceptor groups absorb photons of certain energy and then cause some electrons to jump from the ground state to the excited state. At this time, the molecules in the high-energy excited state will be isomerized by the energy transfer within or between molecules. When a gel polymer binds to a photosensitive functional group, the gel will undergo a reversible color change due to visible or ultraviolet light. The light-responsive gel is induced by light, and the gel has no direct contact with external factors, so remote control can be achieved. It is favored in molecular detection, drug-controlled release system, intelligent light marker, communication equipment, and some other aspects, so it has attracted more and more attention. At present, there are two types of more perfect theoretical systems [59,60].

##### Combining Photosensitive Compounds

Light energy can be converted to thermal energy by photosensitive compounds. Using this mechanism, when exposed to external light, the material heats up to the phase transition temperature and responds. Suzuki and Tanaka [61] used this mechanism to prepare the polymer gel of PNIPAM and the photosensitive compound chlorophyllin (Chlorophyllin). The study found that when the temperature reaches 31.5 °C, with the change of light intensity, the gel undergoes an intermittent volume phase change at a specific moment.

Light decomposition photosensitive compounds will cause ionization. Mamada and Tanaka [62] introduced the Ultraviolet (UV)-sensitive molecules of bis [4-(dime-thy-amino) phenyl] (4-vinyl phenyl)methyl leucamide into the PNIPA gel; the study found that UV irradiation caused the gel to form a large number of ions and present intermittent volume changes.

Photoresponsive polymer gels prepared by combining photosensitive compounds have relatively poor deformation stability and controllable properties, while combining photosensitive groups can hopefully solve this problem better [63].

##### Combining Photosensitive Group

When the photosensitive group is introduced into the gel polymer chain, it will undergo isomerization when it absorbs light. By using this mechanism, the material can respond.

Azobenzene and its derivatives have good environmental stability and easily bond polymers. It exhibits reversible cis–trans isomerism under the action of light, which makes the gel show expansion and contraction. Ultraviolet light can transform trans-azobenzene into cis-azobenzene. Visible light is the opposite. In the process of structural change, the molecular size changes significantly [64,65,66,67].

M. Moniruzzaman and G. F. Finnando et al. [68] prepared a copolymerized gel containing acrylamide and trans-4-methyl propenylhydr oxy azo benzene (MOAB), but the reaction time was too long (about 90 h), and the reaction requirement was harsh. Although the copolymer synthesized by M. Moniruzzaman and G. F. Fernando can respond to ultraviolet light, the study of the photo responsiveness of the copolymer gel needs further research.

To further study the influence of azophenyl groups with photoisomerization on the properties of copolymerized gels, Zhao Yiping et al. [69] adopted the free radical copolymerization method for the first time and utilized the initiation effect of azo-diisobutyronitrile (2,2′-azo-bis (isobutyronitrile), AIBN). A single photoresponsive copolymeric polymer gel P(AAM-co-AAAB) was prepared by the free radical copolymerization reaction of intermediate monomer acrylamide azobenzene (AAAB) with acrylphthalamine (AAm), and N, N1 methylene bisacrylamide (MBAA) was introduced at the same time (as shown in Figure 5). The study found that it is difficult to synthesize 95:5 copolymer gels than 99:1 and 98:2 copolymer gels. The gel strength is low, brittle, and fragile; P (Aam-co-AAAB) gel deformation stability and copolymerization ratio are proportional. Whether in the field of bioengineering or other fields, it has a broader prospect.

Compared with azobenzene, spiropyran is a compound with multiple stimulus responsiveness [70]. Its conversion process is reversible and reproducible. It has been widely used in many aspects (such as the functionalization of gels). The use of spiropyran photochromic mechanism to prepare new light-sensitive gels has been a hot research direction in recent years (as shown in Figure 6).

The traditional synthesis of photosensitive PEG monomer has the disadvantages of complex reaction, cumbersome post-processing, and low productivity. Traditional photosensitive gels have poor mechanical properties and compatibility. Afterwards, many synthetic photoswitch molecules were synthetized to develop new types of materials or to trigger the nanopore aperture. Among them, spiropyran (SP) is particularly interesting because its photoinduced change is not only a simple stereo isomerization but induces a modification of both its chemical structure and physical properties (as shown in Figure 7). Under visible light (VIS), the spiropyran is uncharged and not soluble in water. Under UV irradiation, spiropyran changes to a merocy anine (MC) form, which is zwitterionic and soluble in water [71]. These exceptional properties have been used to functionalize both inorganic and polymer material.

Based on the poor mechanical properties and limited application range of the traditional gels, the new gels have high strength and diversified structure. Yang Quanzhu et al. [72] prepared nitrospiropyran methacrylate (as shown in Figure 8), and copolymerized 0.6% g/mL spiropyran functional body with acrylic acid and acrylamide (the first network monomer) or NIPAM (the second network monomer). Two different spiropyran functionalized photoresponsive hydrogels (SP_1_DN and SP_2_DN gel) were prepared. It was found that both SP_1_DN and SP_2_DN gels showed apparent phase transition behavior, and spiropyran promoted the phase transition behavior of the copolymerization network to some degree. Spiropyran molecules can further regulate the phase transition temperature of the two-network gel in the phase transition behavior. The role of spiropyran molecules in the concept of the two networks provides more possibilities for intelligent soft devices.

##### React Oxide Particles with Hydrogel

Oxide particles have the advantages of high toughness, good stability, and easy preparation [73], but their application in the hydrogel is rarely reported. To study the application of oxide particles in hydrogels, Huang Ke et al. [74] used NIPAM(45 mmol) as a monomer, N, N-methylene bisacrylamide (BIS, molar ratio of 1:10,000) as a chemical crosslinking agent, potassium persulfate (KPS, mass fraction 1.6%) as inducer, tetramethylene diamine (TEMED, mass fraction 1.8%) as an accelerant. Near-infrared response intelligent hydrogel (pure PNIPAM, PNIPAM-GO, and PNIPAM-GO-ZrO_2_) were prepared by chemical crosslinked with 5 mg/mL GO (GO) and 1/3 g/mL ZrO_2_ (as shown in Figure 9, Figure 10 and Figure 11).

It was found that PNIPAM-GO and PNIPAM-GO-ZrO_2_ have good near-infrared response characteristics. Compared with pure PNIPAM, the maximum load quality of PNIPAM-GO and PNIPAM-GO-ZrO_2_ is increased by 1.40 times and 1.70 times, because the ZrO_2_ powder plays a similar role in dispersion strengthening and bear most of the stress. The penetration resistance of PNIPAM-GO-ZrO_2_ hydrogel is significantly improved. The mechanical strength of the near-infrared response intelligent hydrogel prepared by its oxide particles is improved, and it is more widely used in practice.

Due to the light source of energy being not dangerous, with clean environmental protection, easy to control, and can realize non-contact remote control, it has more broad prospects both in the field of daily life and biological engineering field. Combining gel early photographic compounds have light sensitivity, but with the deformation stability and controllability of the gel being not strong, by introducing photosensitive groups (such as azobenzene and spiropyran compounds) into the gel, its deformation stability and controllable performance can be improved effectively. In recent years, some researchers have applied oxide particles with good mechanical properties to hydrogels, and prepared intelligent hydrogels with high mechanical strength and near-infrared light response. Because of its high mechanical strength and having no harm to organisms, its application in the biomedical direction is more meaningful [75,76].

#### 2.1.3. Electric Field Responsive Gel

Macroscopic utilization of nanomaterial provides a new idea for the research and development of novel adsorbent, which can enhance efficiency inthe adsorption and elution process. In this paper, nano-polypyrrole (PPy) was dispersed into two inexpensive and renewable biomass materials, gelatin (Gel) and chitosan (CS), to fabricate a novel photo/electric-sensitive hydrogel, Gel/CS/PPy. The micronetwork of Gel/CS/PPy shows a high adsorption rate of 94.2% for acid fuchsine (AF). Furthermore, with the addition of polypyrrole, Gel/CS/PPy has the characteristic of photo/electric response, which can improve the elution efficiency of AF from the adsorbent [77].

The volume or shape of the electric field responsive gel changes when it is subjected to an applied voltage. There are two general types of electric field responsive gels used in the field of photonic crystals, polyelectrolyte hydrogels, and metal–organic polymer gels. The former is composed of polyelectrolyte absorbed water with dissociable ionized groups in the network, and if it is placed in an electrolyte and a voltage is applied, it will undergo swelling, shrinkage, or bending deformation. Therefore, it can be introduced as an electroactive substance inside the photonic crystal, and the volume of the gel expands or contracts under the action of electric field, pulling the lattice of the photonic crystal to change, thus causing the color of the photonic crystal to change accordingly, and achieving the purpose of electric field response. Zhang Huijie et al. [78] used the monomer hydroxyethyl methacrylate (HEMA) as the backbone and 2-acrylamide-2-methyl propane sulfonic acid (AMPS) as the functional monomer to prepare P(HEMA-co-AMPS) Polyelectrolyte gel (as shown in Figure 12). The study found that with the increase in the proportion of monomer AMPS, the swelling performance of Hema-co-Amps was the best when the amount of crosslinker was 0.5%. The effect of electric field could improve this property of the dry gel, and the gel shape variable of P(HEMA-co-AMPS) reached the maximum value of 85%. The electrosensitive gel photonic crystals were obtained by combining the characteristics of P(HEMA-co-AMPS) (swelling and electrosensitivity) with the characteristics of polystyrene (Polystyrene) photonic crystals (macroscopic color uniformity), which have important applications in sensors and color display. From the infrared spectrum of P(HEMA-co-AMPS) gel, the overlapping peak position of C=O in AMPS and HEMA and the overlapping peak position of S–O in AMPS and HEMA can be obtained. It is obvious to see the overlapping peaks of S–O in AMPS and C–O in HEMA

To prepare environmentally and friendly electric field response gel, Han Xuewu et al. [79] adopted natural gelatin as the raw material, introduced the sulfonic acid polar group in the molecular chain, and synthesized a new sulfonated gelatin hydrogel through a series of reactions. The study found that: under the effect of the applied electric field, the content of the S element was positively correlated with the electric field response-ability of the colloid. When the electric field strength is 1.6 kV/mm, and S content is 1.59%, the responsiveness of the gel is the most obvious. Although the gel prepared from natural gelatin is environmentally and friendly, strong hydrophilic, and biodegradable, it has poor mechanical properties and poor stability.

Hydrogels with good mechanical properties, excellent biocompatibility, and designable shapes are of great importance for their biomedical applications [80]. To develop hydrogels with strong mechanical properties and high stability, and high electrical response, Dai Wenqing et al. [81] selected soluble starch with more hydroxyl groups and different concentrations of FeCl_3_ solution and AlCl_3_ solution as raw materials, and used glutaraldehyde (50%) as the crosslinked agent. Starch-Fe (III) hydrogels (as shown in Figure 13) and starch-Al (III) hydrogels (as shown in Figure 14) were prepared under the conditions of no electric field and different electric field strengths. The study found that: when the content of ferric ion was 0.2 mol/L, the starch-Fe (III) hydrogel showed the strongest response to the electric field. Starch-Al (III) hydrogel has the best response to the electric field when the content of Aluminum ion is 0.15 mol/L. The electrical response of starch-Fe (III) hydrogels and starch-Al (III) hydrogels is better with the increase in electric field intensity. The results showed that starch-Fe (III) hydrogel was better than starch-Al (III) hydrogel. These two gels combine the advantages of strong coordination and polarization of metal ions (Fe, Al). They are complexed with natural high molecular polymers containing active groups to prepare hydrogels, which improves their mechanical and electrical response performance.

Taking advantage of PANI’s high conductivity, low preparation cost, and easy preparation, Shuo Li et al. [82] prepared polyaniline/PAAM (PANI/polyacrylamide) hydrogel (as shown in Figure 15), and further prepared supercapacitor with sandwich structure and portable piezoelectric sensor. The piezoelectric sensitivity of the ECHs wearable sensor and the electrochemical performance of the ECHs supercapacitor were studied. The maximum value of the ECHs supercapacitor was 1022 mF/cm^2^. It was found that the ECHs can carry sensors, and they can sense the subtle changes in body movement. These phenomena indicate that the capacitive and piezoelectric sensitivity of ECHs are closely related to the performance of PANI. PANI can not only endow hydrogel capacitors, but also improve the conductivity of hydrogels. The electro-responsive hydrogels prepared by PANI can well be used to replace inorganic electronic materials and manufacture flexible electronic materials.

Due to its changeable structure, flexible mechanism, and controllable electronic performance, electro-responsive hydrogels can directly convert electrical energy into mechanical response, so electro-responsive hydrogels are widely used in various fields [83]. To increase the mechanical properties of hydrogels, improve the stability of hydrogels, prepare hydrogels with high electrical response performance, and take into account the environment-friendly characteristics, natural polymer materials that are easy to decompose [84,85,86], non-toxic, and renewable (such as starch and gelatin) should be used more. The gel prepared by adding graphite oxide (GO) has high mechanical strength and toughness, which is suitable for the field of bioengineering [87,88].

### 2.2. Chemical Stimulus-Response Gel

#### 2.2.1. pH-Responsive Type

The three-dimensional network structure of such hydrogels usually contains ionizable basic or acidic groups. As the pH value of the medium changes, the groups will ionize, leading to the dissociation of hydrogen bonds between the chain segments of macromolecules in the network, resulting in discontinuous swelling volume changes.

pH-responsive gels have been deeply developed due to their ability to respond to pH changes. Still, their weak mechanical properties and narrow pH-sensitive range have seriously hindered their use [89]. However, its unique response to pH makes it more applicable in drug release systems and bioengineering fields [90,91].

In recent years, the use of biological materials is increasing widely, and the research of cellulose-based composites has become a hot direction. Cellulose (CE) is the most abundant renewable organic material in the biosphere. Cellulose-based adsorbents have been studied in many domestic and international studied in the literature. However, among the reported cellulose-based adsorbents, most of them have low adsorption capacity and are difficult to achieve multiple recycling. Therefore, a highly efficient and multi-cycle regenerable adsorbent is yet to be investigated. Wang Zhicun et al. [92] synthesized pH-responsive CE/SA composite gel by combining Cellulose with Sodium Alginate and using Epichlorohydrin as a crosslinking agent. It was found that it had a good pH response to Methylene Blue (as shown in Figure 16). When the pH value increased from 3.0 to 9.0, the MB removal rate increased from 46.04% to 98.57%; when the pH was greater than 7, the adsorption effect was the best. After 10 cycles of adsorption, the elimination rate of MB remains above 95% (as shown in Figure 17), indicating its excellent recycling reproducibility. The gel has a good effect on MB removal, low cost, and simple synthesis process, and has excellent potential in wastewater treatment and recycling [93].

With indepth research, the dynamic covalent bond has gradually become one of the research hot points. Hou Yifan et al. [94] prepared alginate hydrogel with high elasticity and pH response using polyethylene glycol dibenzoyl hydrazine and alginic acid containing some aldehyde group as raw materials through the characteristics of reversible dynamic covalent hydrazone bond. Under the conditions of hydrogel concentration of 10%, pH = 7.2 and strain of 1%, the variation curves of storage modulus (G′) and loss modulus (G″) with angular velocity (ω) were measured by rheometer (as shown in Figure 18). It can be seen from the figure that G′ > G″, G′ tends to be stable in the high-frequency region, indicating the elasticity and strength of the hydrogel are better. The multi-responsive gel has great potential value in drug delivery, organ repair, and other aspects. At the same time, the hydrogel was prepared from alginate and polyethylene glycol with excellent biocompatibility, so it is expected to be used in injectable cell engineering scaffolds and drug release intelligent materials [94]. Hydrogels harden and become brittle during the swelling process, and, conversely, they become soft and easy to pull. It is found that the elastic modulus of hydrogel is related by the density of molecular chains of hydrogel and the elastic energy Feι contained in each molecular chain, which is expressed as E(Φ) ≈ υ·Feι in the scalar theory.

Biomaterials play an extremely important part in the biomedical field, and the role of medical hydrogel has made it become a hot research direction in the biomedical field in recent years [95,96,97]. Fan Zhiping et al. [98] successfully prepared two precursor macromolecules, PGA-Ty grafted with polyglutamic acid (PGA) tyramine (Ty), and HA-CA grafted with cysteamine (CA) with hyaluronic acid (HA), by using the EDC/NHS method. After the two precursor macromolecules were mixed (as shown in Figure 19 and Figure 20), in the presence of low concentration hydrogen peroxide, the injectable interpenetrating network (IPN) medical hydrogel was successfully prepared by crosslinking with horseradish peroxidase (HRP). Through a series of experiments, it was found that the IPN hydrogel has adjustable mechanical properties, porous internal structure, adjustable degradation properties, and ideal equilibrium water content, and has a good application prospect as a drug carrier material. Its biocompatibility is good. These properties indicate that the interpenetrating network hydrogel system can be used as a candidate material in various medical fields and play a significant role in the biomedical field [99].

The filtration and capture of particles/cells is the basis of single-cell analysis and biomedical research. The microfluidic chip manipulation method has attracted extensive attention because of its low test-dose consumption, low cost, high sample processing efficiency, strong integration, and small volume. Hu Kai et al. [100] added 1.6 g NIPAAM (98%), 0.8 mL AAC (99%), and 0.15 g PVP to 1 mL EL (98%), and fully stirred them to dissolve. Then added 0.1 mL EMK (mass fraction 20%), 0.4 mL DPEPA (mass fraction 98%) and 0.5 mL TEA (mass fraction 99%), stirred thoroughly overnight to obtain a pH-responsive hydrogel. The driving force of pH-responsive hydrogel expansion and contraction comes from the combined effect of electrostatic force and osmotic pressure (as shown in Figure 21). When pH > 9, because the hydrogel polymer network contains AAc chains after photocrosslinking, the carboxyl group (–COOH) on the chain will combine with the hydroxide root (OR) in the solution to become –COO–, thus the chain will accumulate a large amount of On the other hand, because the chain contains a large number of –COO–, the positive ions in the solution will enter the hydrogel in large quantities in order to maintain the electric neutrality, thus leading to a higher concentration of positive ions inside the hydrogel than in the environment outside the hydrogel, and the ion concentration difference caused by osmotic pressure will cause water to enter the inside of the hydrogel. Under the combined effect of electrostatic repulsive force and osmotic pressure, the hydrogel will rapidly absorb water and swell. When pH < 9, –COO– will be protonated and combined with H+ in solution to change back to –COOH, and the electrostatic repulsive force inside the hydrogel will be reduced; meanwhile, in order to maintain the electrically neutral positive ions will return to the solution, and the osmotic pressure will cause water inside the hydrogel to enter the solution outside the hydrogel. The hydrogel will shrink due to the loss of water. Then the hydrogels were processed by using femtosecond laser holographic processing technology and direct writing processing technology, respectively. It was found that the holographic processing technology has more advantages: the processing time is shortened by more than 100 times, the expansion and contraction performance is improved by nearly 20%, and the surface quality is better. The pH-responsive hydrogel microstructure was integrated into the microfluidic chip by femtosecond laser holographic processing technology to achieve multi-particle filtration and complete particle/cell capture. This application provides a technical basis for future single-cell exploration and development. By studying the microscope diagrams and outer diameter data of microstructure expansion and contraction under different laser parameters, it can be obtained that the holographically processed microstructures have better surface quality than single-point processing at a smaller exposure dose with a lower degree of hydrogel cross-linking, which is due to the shorter duration of the holographic process and the less influence of fluid and heat generated perturbations on the structure formation.

IPN hydrogels have been widely used to stabilize nanoparticles and drug delivery, and have attracted significant attention in the scientific community. Shabnam Sattari designed and prepared a hybrid organic–inorganic hydrogel by crosslinked poly (aspartic acid) with graphene nanosheets through the leading network and poly (acrylamide-co-acrylic acid) through the secondary network based on IPN. Applied to curcumin loading and controlled release [101], it was found that curcumin release was maximum and minimum in the gel at pH7.4 and pH2.1, respectively (as shown in Figure 22). Synthetic hydrogels can hinder bacteria, and their loading capacity and controlled release of curcumin make them eligible for future therapeutic applications such as wound dressings.

pH-responsive gels have an obvious reaction to pH value, a wide application range, and easy control of changing conditions. They can make a sensitive response to the change in pH value [102]. However, the poor mechanical properties and limited sensitive range of gels severely limit its practicability. pH-responsive gels made through biological materials or processing technology not only have good biocompatibility, but also have the dual properties of pH stimulation response and high mechanical strength. Both in the industry and other fields have a broader development prospect [103,104].

#### 2.2.2. Chemical Response Type

The swelling behavior of this responsive gel will change suddenly because of the stimulation of special chemical substances (such as sugar) [105]. Glucose responsive gels are highly favored by researchers in the fields of glucose detection, controlled release of drugs, integration of diagnosis and treatment, software robotics, etc., and have been widely used to construct intelligent feedback system [102,106]. Glucose responsive microgels can be prepared by controlling the particle diameter at the micron level, which is different from the bulk and is expected to broaden its application scope in biosensing, engineering, and other fields [107].

Yao Dan et al. [108] synthesized PLGA (polylactic acid-co-hydroxyacetic acid) polymer by ring-opening polymerization of ethyl lactide and lactide, and prepared PLGA porous microspheres by selecting bovine serum protein (BSA), and then introduced PBA group on the surface and combined with cis-dihydroxyl to obtain two kinds of composite gel: PLGA porous microspheres/polyvinyl alcohol (PVA), HA-DOPA (dopamine modified hyaluronic acid)/PLGA porous microspheres. The functional groups hydroxyl (–OH), methyl (–CH_3_) and methylene (–CH_2_) carbonyl (–C=O), and ether bond (–C–O–C) are important components in PLGA. The study found that: they were both sensitive to glucose and capable of physiologically self-regulating long-term insulin release. It provides a new scheme for the development of an insulin physiologically self-tuning implantable gel delivery system with application prospects for diabetic patients. Moreover, it provides a good reference for the development and exploration of other drug delivery systems.

Glucose (GLU) responsive gel fibers are very sensitive to changes in external glucose concentration. They have a fast response rate, which has excellent potential for development in drug release, bioengineering, protein carrier, and other fields [109,110]. Ma Yuanyuan et al. [111] prepared Clay physical and BIS chemical crosslinked continuous glucose-sensitive P(NIPAM-Co-AAPBA) gel fibers by self-made T-type microfluidic chip and using photoinitiated free radical polymerization and microfluidic real-time spinning method. It was found that SPNA(B) reached a stationary state in 15 min, which only accounted for 2.5% of the equilibrium time (600 min) of the same proportion of bulk gel. It indicates that the increase in glucose response speed is due to the decrease in gel fiber. However, SPNA(C) reached equilibrium at 12 min, compared with SPNA(B) at 15 min, indicating that SPNA(C) had a faster glucose response speed (as shown in Figure 23 and Figure 24). The research of glucose-sensitive hydrogels using gel fibers to monitor and respond to GLU content has attracted significant attention in bioengineering.

Real-time monitoring of blood glucose levels can reduce other symptoms caused by diabetes, which is beneficial to patients’ health. Wang Jun et al. [112] prepared PBA-based microgel Poly(AAPBA-DMAA-co-AAm) and added CS and corresponding monomer (AAPBA, DMAA, AAm) to obtain IPN hydrogel (as shown in Figure 25). When no monomer (AAPBA, DMAA, AAm) was added, IPN hydrogels named Poly (AAPBA-DMAA-co-AAm)/CS, based on CS interpermeable microgel Poly (AAPBA-DMAA-co-AAm) CS were prepared and named Poly (AAPBA-DMAA-co-AAm)-CS (as shown in Figure 26). They were sensitive to changes in glucose content, and the maximum physiological pH swelling was 6.2%. The response to glucose at physiological pH reached 7.5%. When CS content was 5%, microgel was applied for 5 h, adding ratio was 100% (volume fraction), the response of Poly (AAPBA-DMAA-co-AAm)-CS hydrogel reached the highest of 23% (as shown in Figure 27), and swelling time was 25 min. Compared with the IPN hydrogel based on PBA copolymer microgel, the response degree increased two times. It provides a more efficient way to measure blood sugar levels.

Designing and constructing a system with sustainable output feedback and real-time detection can use glucose-responsive polymer microgels. Lan Ruyue et al. [113] modified the preparation route of s-lectin, synthesized s-Lectin, and encapsulated it in the PNIPAM network to obtain a novel glucose-sensitive microgel. The results showed that the swelling degree of microgel was proportional to the glucose content (as shown in Figure 28a). Glucose was removed by dialysis, and added glucose again. It was found that glucose had good reversibility in response to the volume phase transition, and the microgel had good stability (as shown in Figure 28b). In the absence of sugar, the phase transformation temperature of the lectin microgel was about 27.3 °C. With the increase in glucose concentration to 10.0, 20.0, and 30.0 mmol/L, the phase transformation temperature increased to 28.6, 29.9, and 30.7 °C, respectively. This confirms the reasonability of the glucose response mechanism (as shown in Figure 28c,d). Using pig blood serum as a complex biological system model to test the high selectivity of quantitative microgels for glucose detection, is expected to be used in the direction of blood glucose detection. A novel glucose-based microgel with high selectivity for glucose recognition may be further integrated with optical properties for visualizing and sustainably testing glucose and output feedback systems.

Contractile glucose-sensitive microgels, which contract when glucose is added, have potential applications in self-regulating insulin release and glucose sensing. Tang, Z et al. [114] synthesized contractions glucose-sensitive microgel P(NIPAM-2-AAPBA) with 2-aminoborate P(NIPAM-AAC) microgel under the catalytic action of EDC. A new mechanism of glucose absorption was proposed: glucose increased the temperature of microgel dispersion, resulting in the contraction of microgel. The combination of glucose and PBA group shortened the distance between glucose and the PNIPAM chain, and significantly enhanced the ability of glucose to reduce VPTT of microgel. According to this new mechanism, the VPTT of the microgel can be adjusted to control the optimal temperature of glucose sensing. Under the guidance of this new mechanism, a novel contractile glucose-sensitive microgel operating at physiological temperature was successfully synthesized. It is expected to be used in glucose sensing direction.

Chemical-responsive gels can change rapidly due to the stimulation of specific chemicals [115], which glucose is the most studied. Glucose-responsive gels are sensitive to changes in glucose concentration. They can monitor and respond to glucose concentration in vivo, to realize intelligent monitoring of glucose concentration in diabetic patients. Therefore, there is a great space for development in the research and application of insulin release systems and biosensors, which needs further exploration.

## 3. Multiple Stimulus-Response Type

Multi-response intelligent gels respond to more than two environmental factors [32].

Intelligent polymer gels can respond intelligibly to external stimuli, showing broad development potential in drug-controlled release systems, memory switches, material separation, and other fields. However, with the increasing improvement and further development of single responsive polymer gels, the development of good biocompatibility and multiple responses, especially the gels that can respond to two or more external stimuli at the same time, have gradually become dominant [116,117].

A tri-responsive and fast self-healing organogel based on boronate ester bond, acylhydrazone bond and disulfide bond was prepared from polyvinyl alcohol (PVA), 4-formyl phenyl boronic acid (FPBA), and 3,3′–dithiobis (propionohydrazide) (DPH). PVA was selected as gel skeleton to improve the mechanical property of the organogel. Tensile test indicated that the obtained organogel is highly stretchable. Due to the dynamic nature of reversible covalent chemistry, the organogel can respond to pH, glucose and redox to undergo gel-sol phase transition. Microscopy observation and tensile test further indicate that the organogel can self-repair without external stimuli [118].

To prepare dual-sensitive injectable hydrogels, Meng Fandong [119] and others used the materials BLG and PLG, the initiator macromolecule PEG44-NH_2_, and obtained a series of block copolymers PEG-b-(PPLGm-co-PBLGn) through the ring-opening polymerization of nitrogen-carboxyl internal acid anhydride (NCA)/PEG-b-PPLGm-b-PBLGn. Through processing, a pH and temperature-responsive hydrogel is obtained. It was found that the strain increased to 10%, the storage modulus decreased rapidly, and the hydrogel properties were weak; the strength increased with the temperature, at 40 °C, the storage modulus and loss modulus were both Rises from 10^2^ Pa to 2 × 10^3^ Pa and 50 Pa to 370 Pa, respectively (as shown in Figure 29). Through the comparison of adding NaCl and CaCl_2_, it was found that the strength of the gel with CaCl_2_ was stronger than that with NaCl, and the ability to resist strain was stronger. The gel’s responsiveness makes it more suitable for medical applications.

To develop new characteristics of intelligent injectable hydrogels, Chen Yan et al. [120] prepared three different temperature/pH response polymers using the RAFT (Reversible Addition-Fragmentation Chain Transfer Polymerization) method: LCST and anionic pH response triblock NDA (PNIPAM-b-PDMA-b-PAA), LCST and cationic pH response triblock NDD (PNIPAM-b-PDMA-b-PDMAEMA), UCST (upper critical solution temperature), and anionic pH response triblock ADAA (P(AM-co-AN)-b-PDMA-b-PAA). The results showed that NDA could achieve reversible sol-gel conversion when the concentration was greater than 7 wt%, pH5 and 37 °C. MTT(3-(4,5)-dimethylthiahiazo (-z-y1)-3,5-di-phenytetrazoliumromide, MTT method, also known as MTT colorimetric method, is a method to detect cell survival and growth) experiments showed that the survival rate of HELF cells was about 90%. The sol-gel reversible transformation of NDD was realized at concentration of more than 15 wt%, pH11 and 55 °C. MTT toxicity test showed that the survival rate of HELF cells was more than 85%. The sol-gel reversible transformation of ADAA was realized at concentration of more than 9 wt%, pH7 and 38 °C (as shown in Figure 30 and Figure 31). MTT toxicity test showed that the survival rate of HELF cells ranged from 82% to 92% (as shown in Figure 32). Overall, ADAA has significant biocompatibility and has great advantages in the medical field.

Shouxin Liu et al. [121] synthesized a temperature- and pH-responsive ABC triblock copolymer MPEG-b-P(MEO_2_MA-co-HMAM)-b-PDEAEMA (PMHD) by RAFT. It was found that the LCST of PMHD could be precisely regulated to physiological temperature (37 °C) by controlling the feed ratio of MEO_2_MA and HMAM (Human mammaglobin copolymer). PMHD4 weak hydrogel had a significant sustained-release effect on hydrolyzed bovine serum albumin drugs in vitro (as shown in Figure 33 and Figure 34). Therefore, hydrogels are a promising candidate for the control of protein drug delivery, and the hydrogel system can be considered as a promising candidate system for the continuous administration of hydrophilic drugs.

There are few reports of injectable hydrogels with multiple stimulus responses to electric fields and pH as drug delivery systems. Jin Qu et al. [122] synthesized an antibacterial conductive injection hydrogel by using chitosan grafted polyaniline (CP) copolymer and xyloglucan (OD) as crosslinked agents (as shown in Figure 35). The release rates of model drugs amoxicillin and ibuprofen in CP/OD hydrogels were significantly increased with the increase in voltage. The conductivity of CP/OD gel was increased to 0.079 s/m (as shown in Figure 36). The toxicity test with cell L929 proved its good compatibility (as shown in Figure 37). Taken together, these injectable pH-sensitive conductive gels with antibacterial activity may be an ideal candidate for intelligent delivery vectors for precise doses of drugs to meet actual needs [101].

Stimulus-response polypeptide hydrogels have attracted attention due to their biodegradation, biocompatibility, and responsiveness to external stimuli [123]. Zhao S, Shuai S et al. [124] successfully synthesized MPEG1K-PLYSY-PVY hydrogel by a simple and economical method, and its gelation temperature (TG) covered the physical temperature of 37 °C. Approximately 36% and 100% of doxorubicin (DOX) were released within 48 h at pH 7.4 and 5.7, when doxorubicin was loaded into the hydrogel (as shown in Figure 38). The rapid release capacity of the gel system in acidic environments suggests that the system may be suitable for drug targeted release. Such biodegradable polypeptide hydrogels with temperature and pH sensitivity may become a new generation of intelligent materials with excellent development potential in the biomedical field [125].

Dual-sensitive intelligent gels have good biocompatibility, biodegradability, and multiple response functions, and have excellent development potential in biomedical and other directions. They have more advantages than single-responsive polymer gels, and can be applied to more fields and cope with more complex situations. However, due to various limitations, they have not yet been used on a large scale [126]. It is suggested to further study and clarify the sensitive mechanism, improve the mechanical strength and response speed of gels through the coordination of different technologies, and construct environmentally sensitive gels with good compatibility and biodegradability from the perspective of bionics [127] to promote its industrial and large-scale production vigorously.

With the deepening of the research, temperature, pH, light, electric field, the chemical sensitive gel has many kinds of research and development, their mechanical strength and response performance are improved at the same time, some areas of demand are raised, they not only require high mechanical strength and good response performance of a single; at the same time they also require good biocompatibility, biodegradability, and function diversity, From this, the research and preparation of multi-response gels are derived from meeting the high demand in the field.

Both single-response and multi-response intelligence polymer gels have their advantages, but also have defects. The current comparison is shown in Table 1.

## 4. Outlook

At present, the research on intelligent polymer gel is still insufficient, which requires systematic study and large-scale application. In the future research, attention should be paid to the development and application of the following aspects: 1. synthesis and development of intelligent polymer gels with high mechanical strength and toughness and self-healing function [128]; Santidan Biswas et al. [129] in the United States designed a polymer gel containing concealed sites and lower critical solution temperature (LCST). This gel can prevent damage by utilizing the concealed sites with the mechanical response and self-healing function. The automatic route of the gel, which dynamically adjusts through deformation to change the behavior of the system, does not require high temperature or extreme conditions, and is energy-efficient and cost-effective. It also responds to deformation in a self-reinforcing or self-reinforcing manner, allowing materials designed with the gel to have a longer service life. The gel can effectively advance the common demand for materials with high mechanical stress requirements, and can be used in the emerging field of mechanically changing materials. 2. At present, the high-strength intelligent polymer gels are not diversified enough. More intelligent polymer gels with various functional requirements should be developed and prepared to meet the needs of more fields; Manjula et al.in Portugal developed a PVA magnetic response hydrogel doped with Iron oxide nanoparticles (MNPS), and found that the magnetic behavior of these hydrogels affected the adsorption of proteins. The magnetic field reduces its ability to adsorb proteins while stimulating it to release them. Therefore, these hydrogels offer the ability to control protein content through cyclic changes in magnetic field strength, and the ability to magnetically modulate protein adsorption makes them promising for developing functional devices for tissue engineering, drug delivery applications, or biosensor systems. 3. In the future, we should research and develop intelligent polymer gels that have the function of biological tissue and even surpass the function of biological tissue. Chang Yanjiao et al. [130] prepared poly N, N-dimethyl acrylamide intelligence hydrogel by physical crosslinked method, using monomer: N, N-dimethyl acrylamide, crosslinked agent: lithium magnesium silicate, reinforcing phase: nano-wood pulp cellulose and ferric oxide as magnetic response particles. Bionic magnetic response 4D intelligent hydrogel with swelling and magnetic response deformation characteristics was successfully prepared by “one-step” mold forming technology. The mechanical strength and multi-form intelligent response can be taken into account effectively, which provides an effective new idea and method of bionics for solving the bottleneck problem of magnetic response intelligent hydrogel.

In the future, with the continuous deepening in research by scientific researchers, it is believed that there will be more types of polymer gels with better performance in various fields.

## 5. Conclusions

As a popular polymer material, intelligent polymer gel has great potential in drug engineering, molecular devices, biomedicine, and other fields. Its development provides essential significance for realizing intelligent soft materials and environmental friendliness. This paper draws the following conclusions based on the current progress.

(1)Single-response smart polymer gels have a single response and limited function, and generally have disadvantages such as poor biocompatibility and insufficient mechanical properties. They gradually cannot meet the needs of the development of various fields, and most of the current research is on physical response gels, with relatively little research on chemical response gels, and the research is still at the basic stage.(2)The multi-response intelligent polymer gel can respond to multiple external stimuli, and combines various single response characteristics. Yet these properties are not so directly connected, some of multi-response gels possess poor biocompatibility and no biodegradability, while gels with no responsive qualities may possess both of the abovementioned qualities. Some multi-response gels also have high mechanical strength and self-healing ability, which are more suitable for higher requirements in specific fields, but the diversification of functionality also means that their preparation methods are more complicated, and large-scale applications are not yet available, and still focus on a single response.(3)At present, most intelligent polymer gels have complex synthetic routes, poor biological compatibility, few response types, and poor mechanical properties. It is suggested to strengthen the development of these four characteristics to prepare new, environmentally, friendly, and excellent intelligent polymer gels, comparable to or even better than biological tissues. Temperature-sensitive smart hydrogels are commonly used as drug-delivery systems and show a great potential as responsive systems for cell detachment. Cytotoxicity is the main concern when using this type of stimuli-responsive hydrogel in the biomedical field. Magnetic hydrogels capable of changing their shape and properties in response to an external magnetic field have attracted high interest as a result of their numerous applications in the medical field. In the future, multi-responsive hydrogels will be the the next generation of smart hydrogels because they can respond to different stimuli and mimic biological processes.

## Figures and Tables

**Figure 1 molecules-28-04246-f001:**
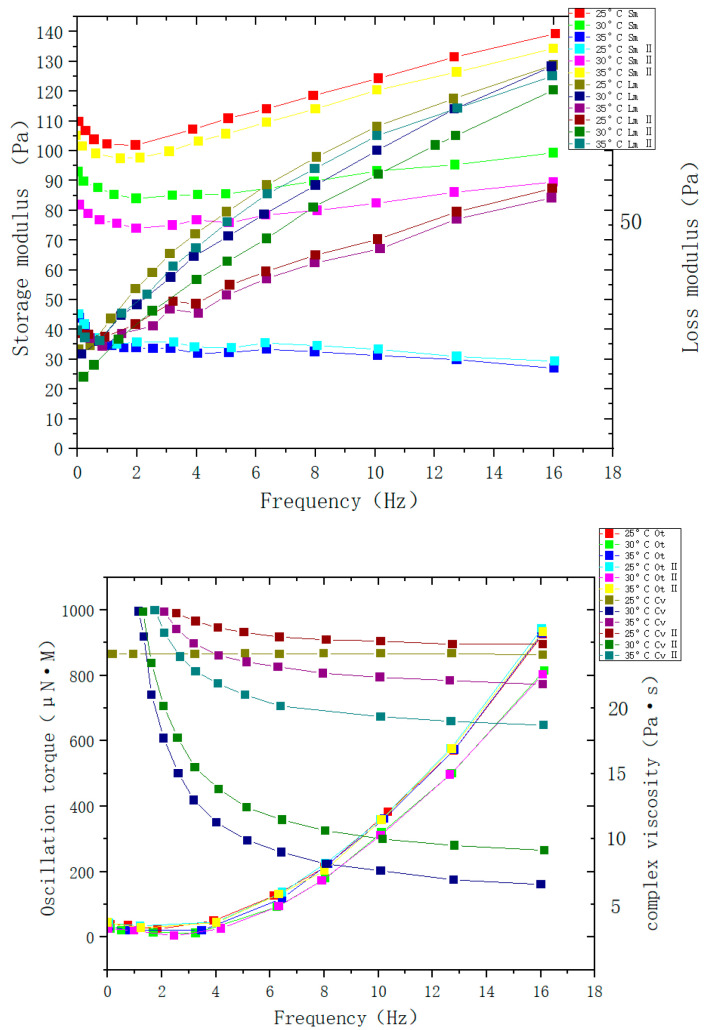
Mechanical properties of temperature-sensitive hydrogels (adapted from [14]).

**Figure 2 molecules-28-04246-f002:**
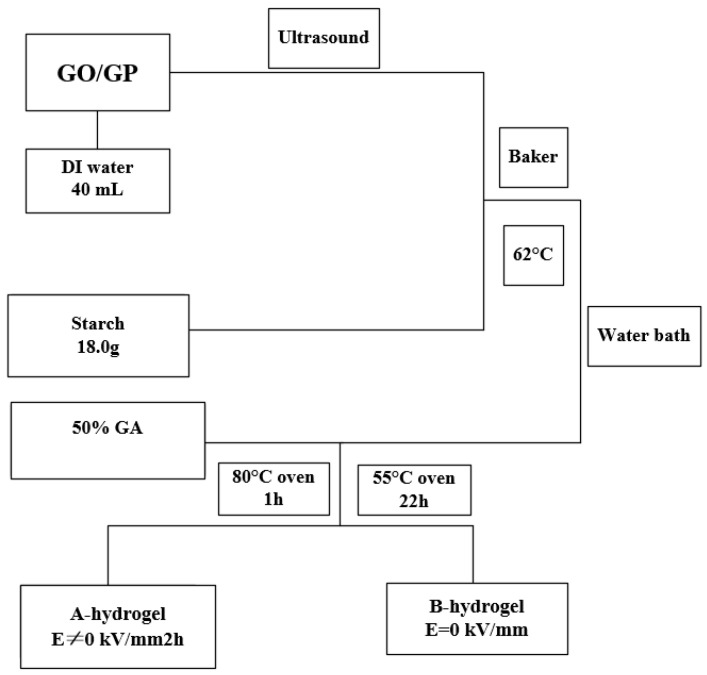
Preparation process of GO/starch hydrogel (adapted from [18]).

**Figure 3 molecules-28-04246-f003:**
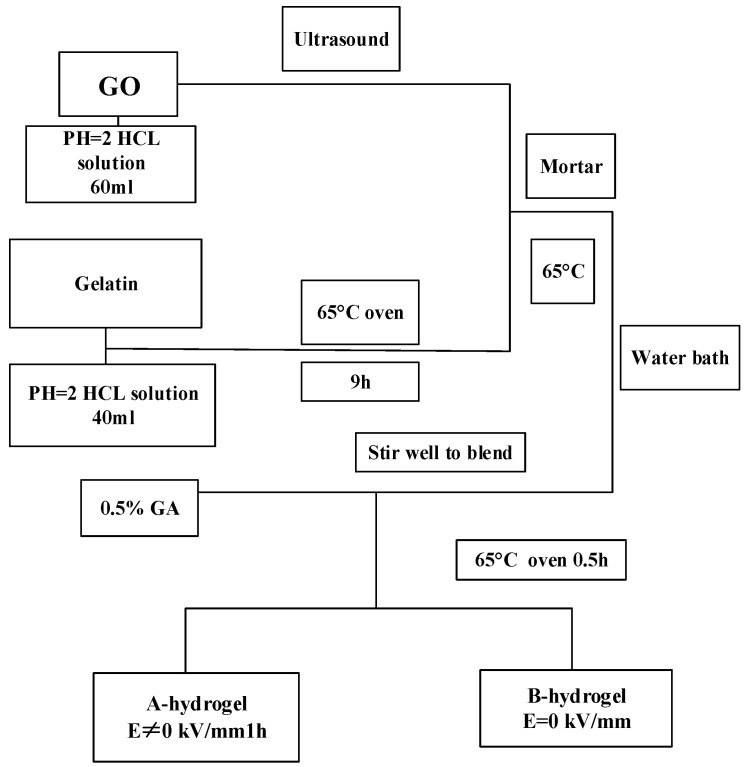
Preparation process of GO/gelatin hydrogel (adapted from [18]).

**Figure 4 molecules-28-04246-f004:**
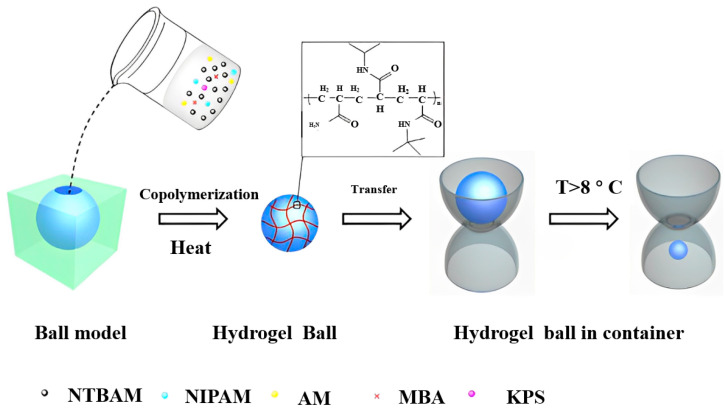
Synthesis scheme of low-temperature responsive hydrogel sphere and temperature monitoring vessel “Reproduced with permission from Tang L. et al., Design of low temperature-responsive hydrogels used as a tem-perature indicator, Polymer, 2019” [56].

**Figure 5 molecules-28-04246-f005:**
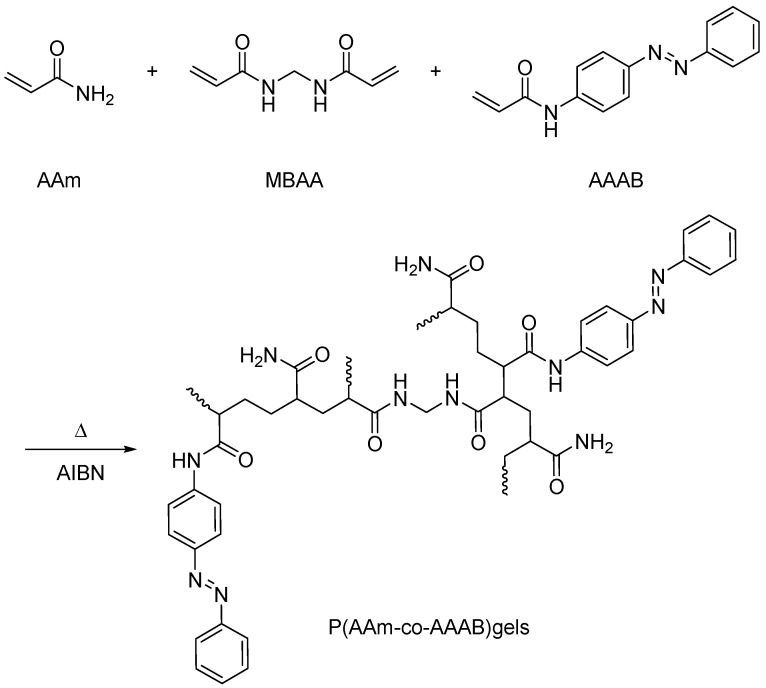
Synthesis of photoresponsive P(AAM-co-AAAB) copolymer gel (adapted from [69]).

**Figure 6 molecules-28-04246-f006:**
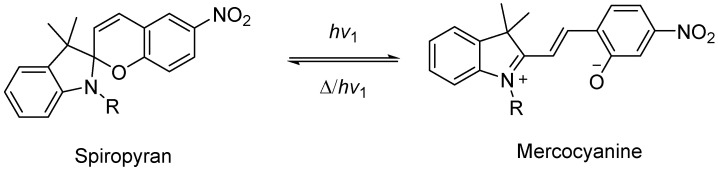
Photochromism mechanism of spiropyran (adapted from [70]).

**Figure 7 molecules-28-04246-f007:**
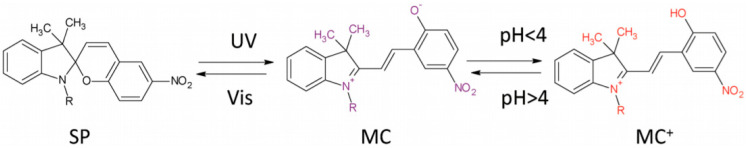
Chemical structure of spiropyran “Reproduced with permission from Ma, T., et al., Combining Light-Gated and pH-Responsive Nanopore Based on PEG-Spiropyran Functionalization. Adv. Mater. Interfaces 2018” [71].

**Figure 8 molecules-28-04246-f008:**
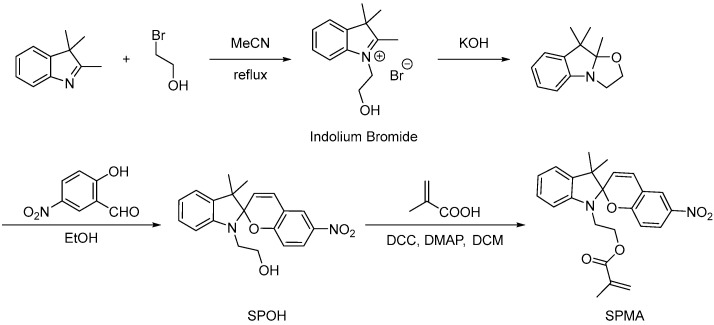
Schematic diagram of synthesis and modification of nitrospirofuran monomer (adapted from [72]).

**Figure 9 molecules-28-04246-f009:**
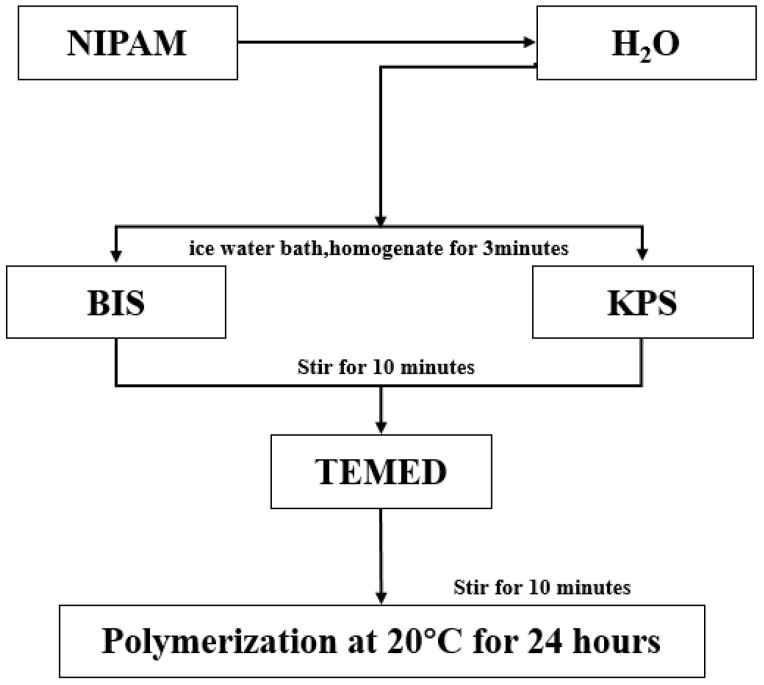
Flowchart of preparation of pure PNIPAM hydrogel (adapted from [74]).

**Figure 10 molecules-28-04246-f010:**
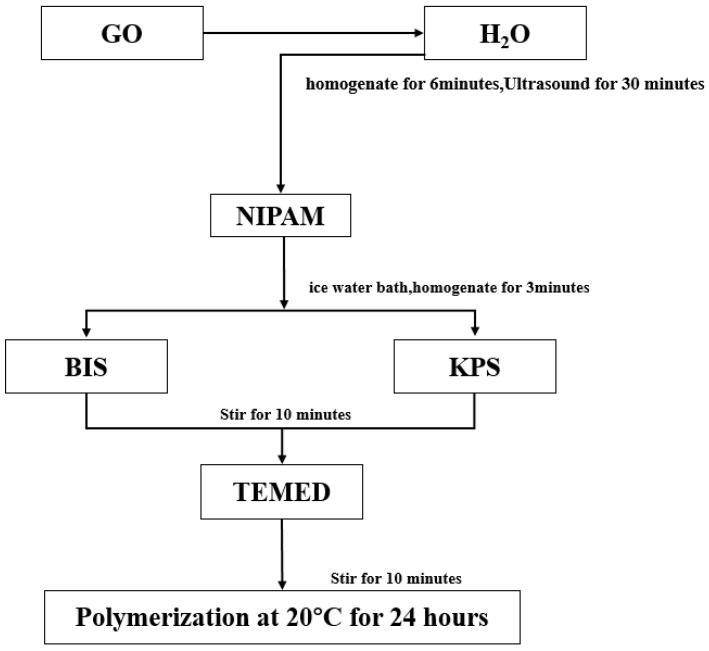
Flowchart of preparation of PNIpam-Go composite hydrogel (adapted from [74]).

**Figure 11 molecules-28-04246-f011:**
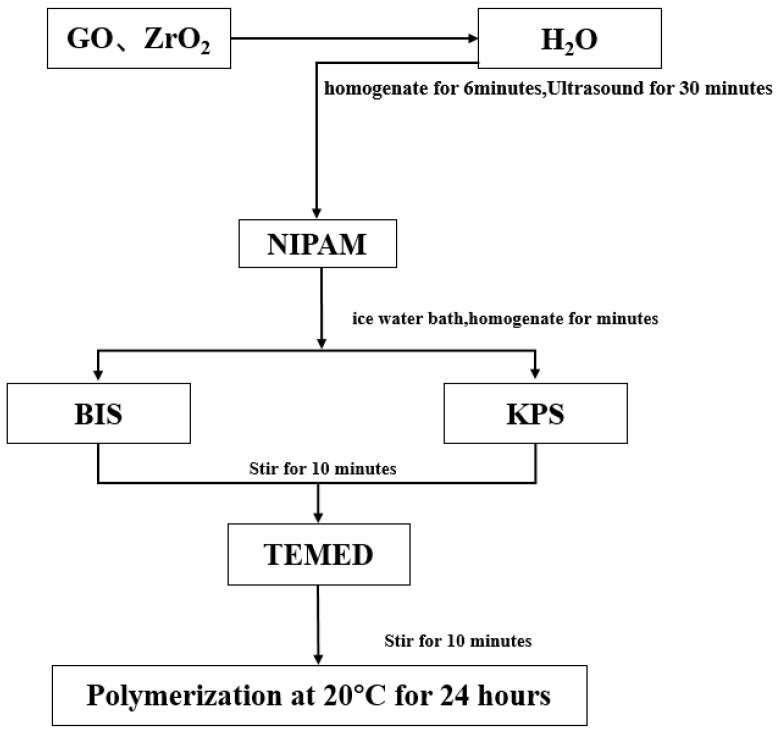
Flowchart of preparation of PNIpam-Go-ZRo_2_ composite hydrogel (adapted from [74]).

**Figure 12 molecules-28-04246-f012:**
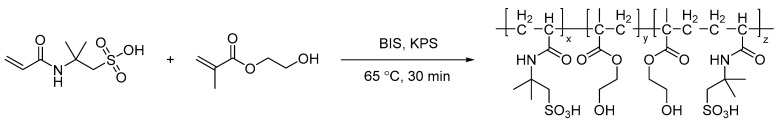
Synthesis roadmap of P(Hema-co-AmPS) gel (adapted from [78]).

**Figure 13 molecules-28-04246-f013:**
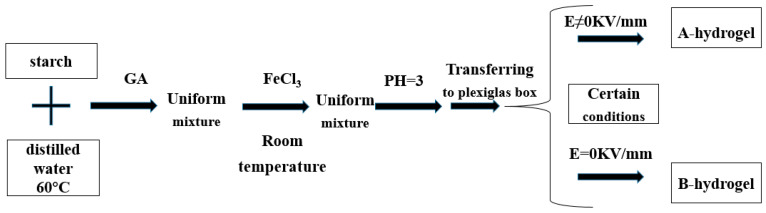
The preparation process of starch-Fe (III) hydrogel (adapted from [81]).

**Figure 14 molecules-28-04246-f014:**
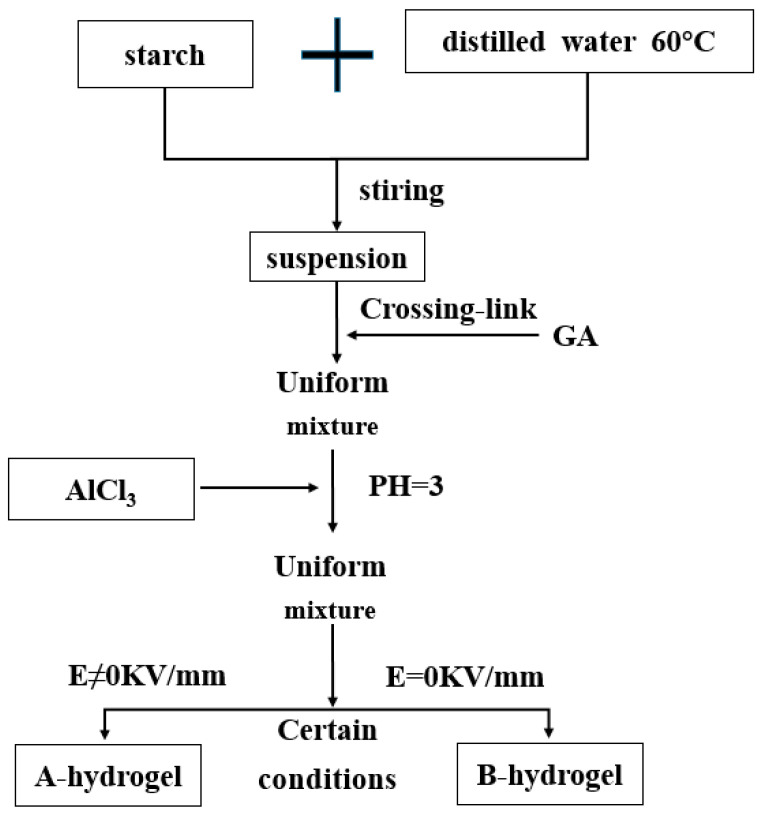
The preparation process of starch-Al (III) hydrogel (adapted from [81]).

**Figure 15 molecules-28-04246-f015:**
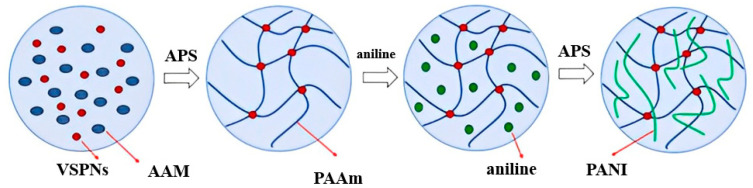
Synthesis process of conductive polyaniline/PAAM hydrogel “Reproduced with permission from Shou L., et al., Bifunctional polyaniline electroconductive hy-drogels with applications in supercapacitor and wearable strain sensors, Journal of Biomaterials Science Polymer Edition, 2020(16)” [82].

**Figure 16 molecules-28-04246-f016:**
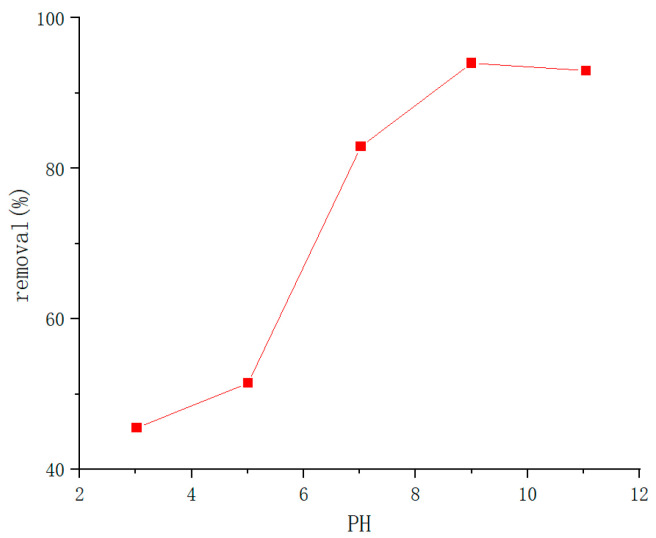
Effect of pH on dye removal efficiency (adapted from [92]).

**Figure 17 molecules-28-04246-f017:**
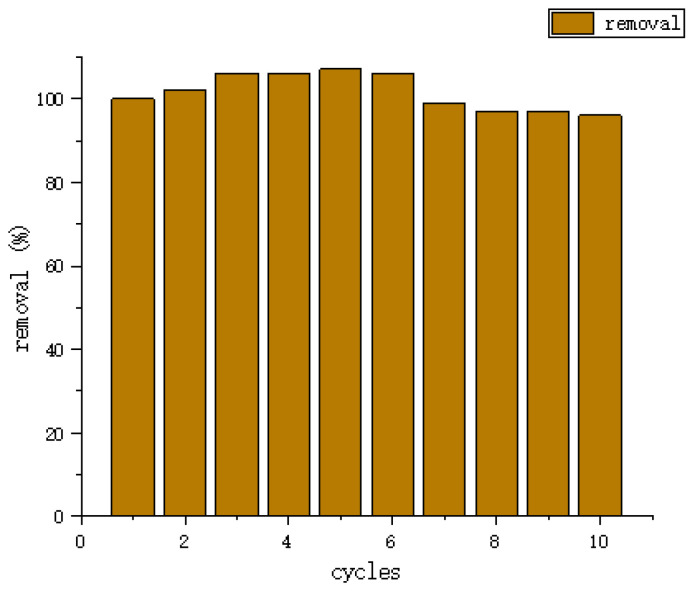
The cycle number of CE/SA composite gel for MB (adapted from [92]).

**Figure 18 molecules-28-04246-f018:**
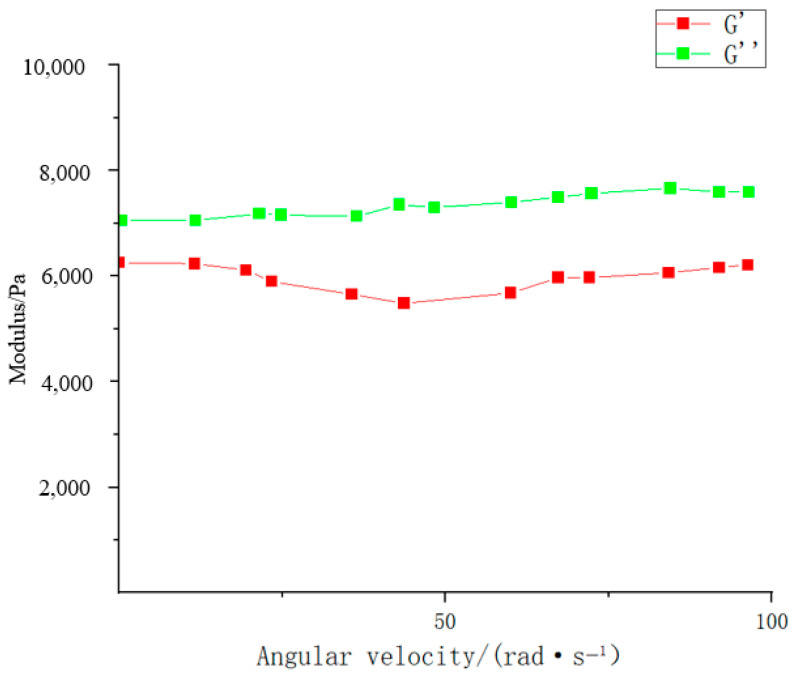
Schematic diagram of rheological properties of hydrogel [94] (adapted from Preparation of alginic acid hydrogel based on reversible covalent acylhydrazone bond and its pH responsiveness).

**Figure 19 molecules-28-04246-f019:**
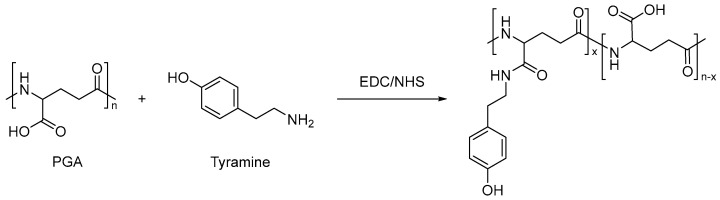
Synthesis of precursor macromolecule PGA−Ty (adapted from [98]).

**Figure 20 molecules-28-04246-f020:**
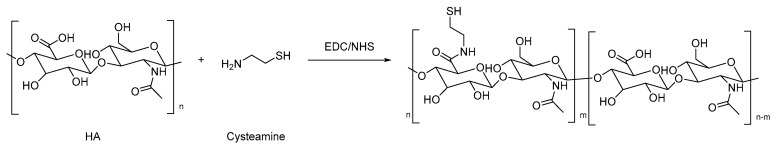
Synthesis of precursor macromolecule HA−CA (adapted from [98]).

**Figure 21 molecules-28-04246-f021:**
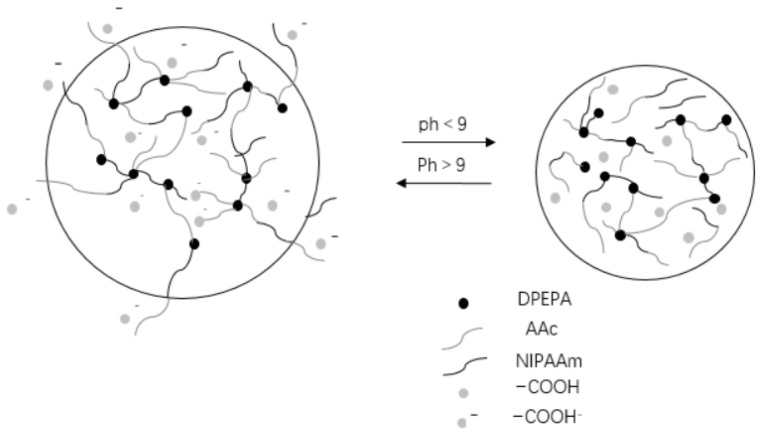
pH response mechanism of hydrogel response (adapted from [100]).

**Figure 22 molecules-28-04246-f022:**
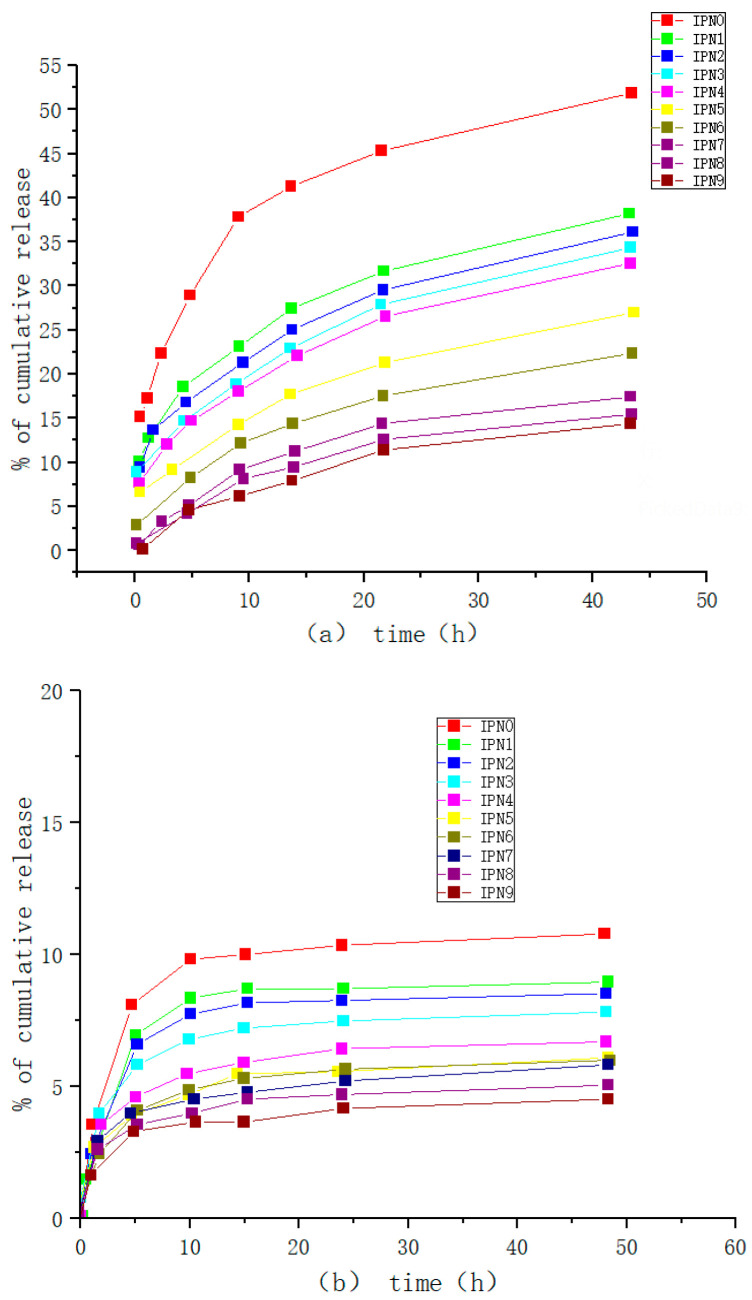
Release profile of curcumin from IPNs (**a**) pH 7.4 and (**b**) pH 2.1 (adapted from [101]).

**Figure 23 molecules-28-04246-f023:**
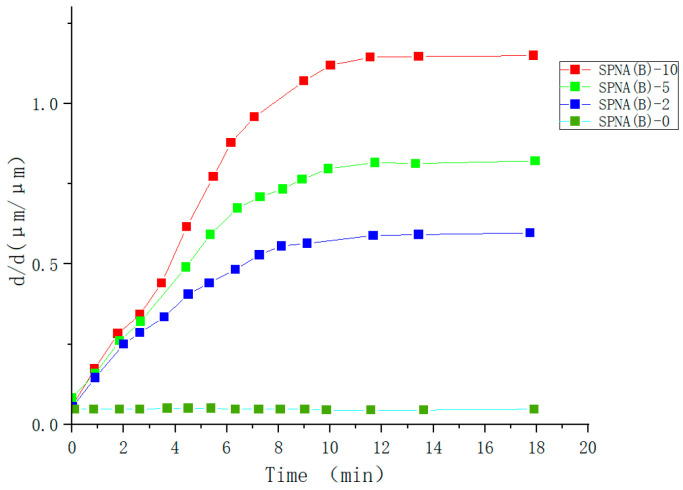
Glucose sensitivity of hydrogel fibers by BIS cross-linking in 400 mg/d L glucose solution (adapted from [111]).

**Figure 24 molecules-28-04246-f024:**
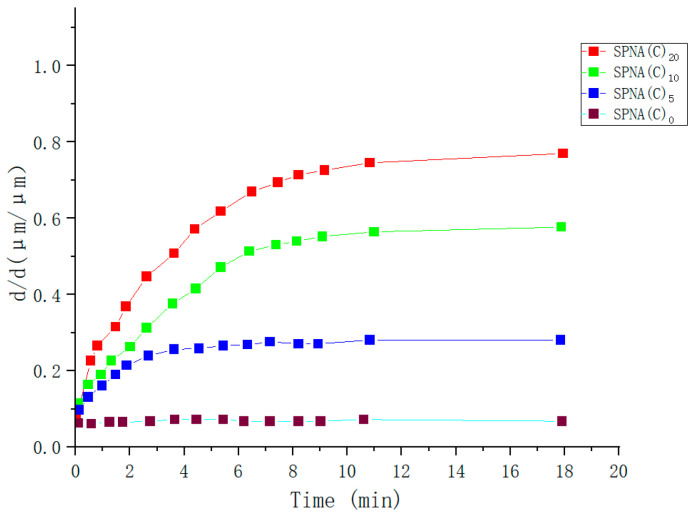
Glucose sensitivity of hydrogel fibers by Clay cross-linkingin 400 mg/d L glucose solution (adapted from [111]).

**Figure 25 molecules-28-04246-f025:**
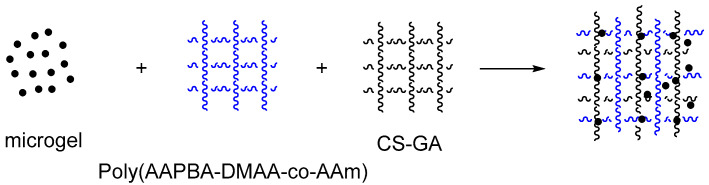
The diagram of double interpenetrating networks Poly(AAPBA-DMAA-co-AAm)/Poly(AAPBA-DMAA-co-AAm)-CS hydrogels (adapted from [112]).

**Figure 26 molecules-28-04246-f026:**
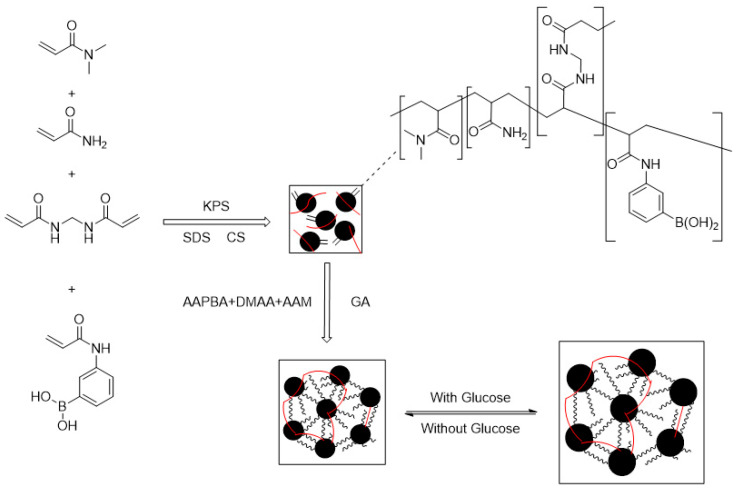
IPN Poly(AAPBA-DMAA-co-AAm)CS/Poly(AAPBA-DMAA-co-AAm) hydrogels synthesis schematic diagram (adapted from [112]).

**Figure 27 molecules-28-04246-f027:**
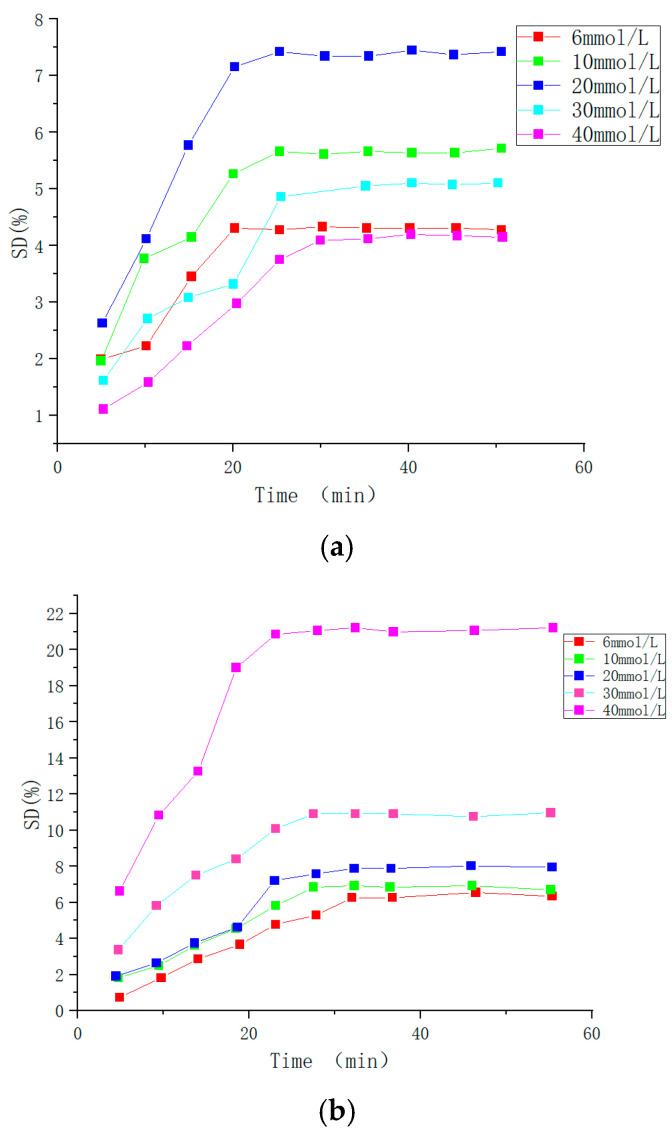
The SD change in IPN hydrogels at different glucose concentration. (**a**): Hydrogel Poly (AAPBA-DMAA-co-AAm)/CS based on PBA copolymer microgel; (**b**): Hydrogel based on CS interpenetrating network microgel Poly (AAPBA-DMAA-co-AAm)-CS (adapted from [112]).

**Figure 28 molecules-28-04246-f028:**
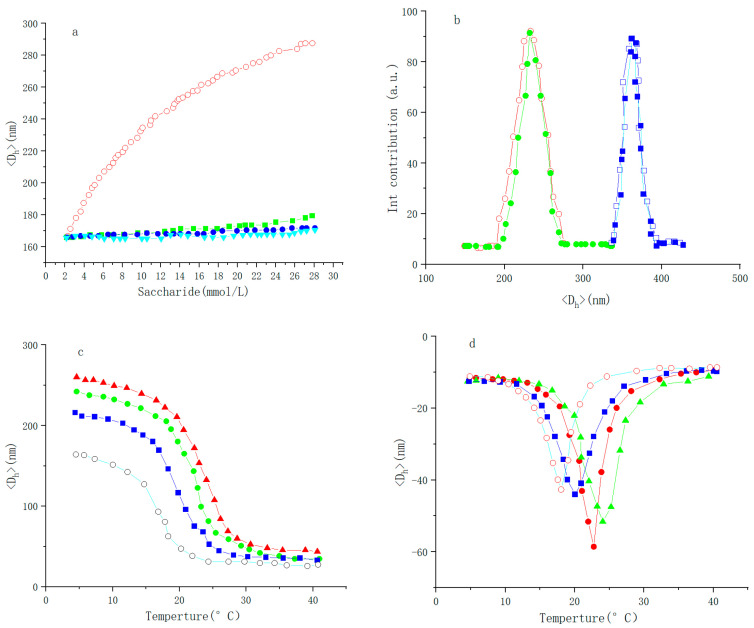
(**a**) The <D_h_> of the microgels in the presence of glucose (○), fructose (■), mannose (●), or galactose (▲), measured at 25.0 °C; (**b**) DLS size distribution of the microgels upon repeated adding (□, ■: 30.0 mmol/L) and removal of glucose (○,●: 0 mmol/L), measured at 25.0 °C; (**c**) Temperature-dependent <D_h_> of the microgels in the presence of glucose (○: 0 mmol/L; ■: 10 mmol/L; ●: 20 mmol/L; ▲: 30 mmol/L); (**d**) The first derivative of <D_h_> with respect to temperature (glucose (○), fructose (■), mannose (●), or galactose (▲)) (lines are asymmetric Lorentz fits. All measurements were made on 0.050 wt% microgel dispersion in 5.0 mmol/L PBS of pH = 7.4) (adapted from [113]).

**Figure 29 molecules-28-04246-f029:**
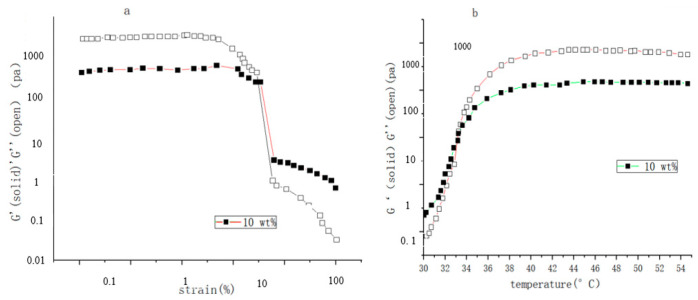
(**a**) The strength curves of storage modulus (G′) and loss modulus (G″) in the gel state changing with strain; (**b**) the curve of the strength of the gel’s storage modulus (G′) and loss modulus (G″) with the change in temperature at pH3.3 (adapted from [119]).

**Figure 30 molecules-28-04246-f030:**
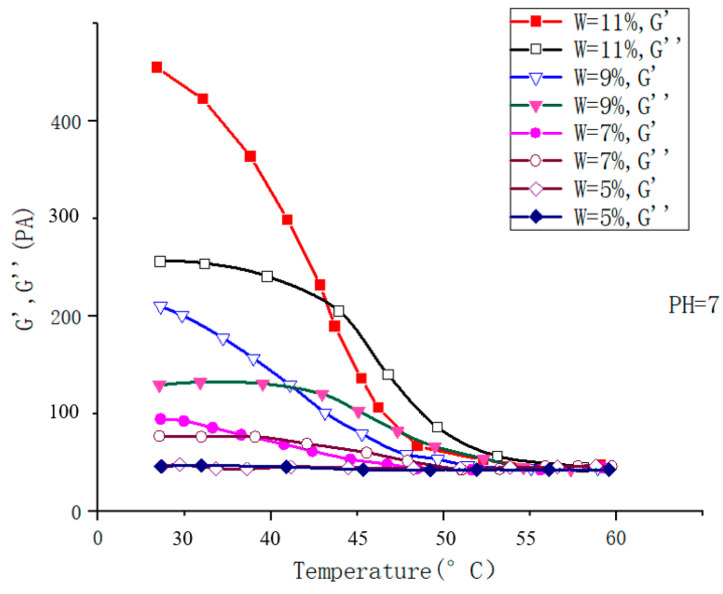
Rheological curves: Concentration-dependent dynamic shear moduli (G′ and G″) for ADAA (pH = 7) with different temperature (adapted from [120]).

**Figure 31 molecules-28-04246-f031:**
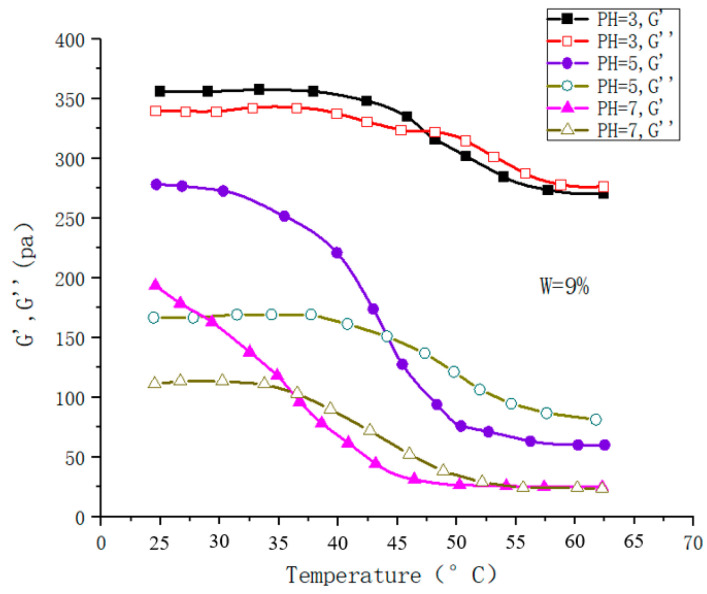
Rheological curves: pH-dependent dynamic shear moduli (G′ and G″) for ADAA (w = 9%) with different temperature (adapted from [120]).

**Figure 32 molecules-28-04246-f032:**
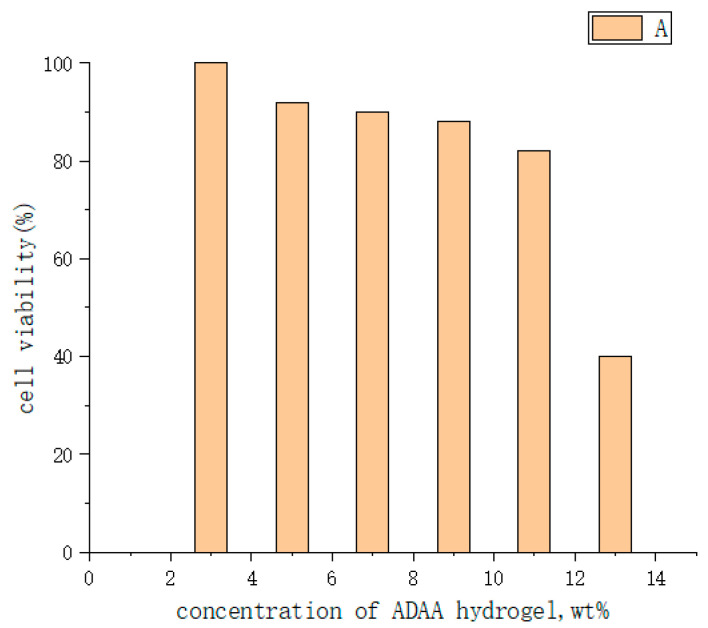
In vitro cytotoxicity of the ADAA hydrogel after incubation with cells for 48 h at 37 °C. (**A**) 37 °C, pH = 7; (**B**) 37 °C, 9 wt% (adapted from [120]).

**Figure 33 molecules-28-04246-f033:**
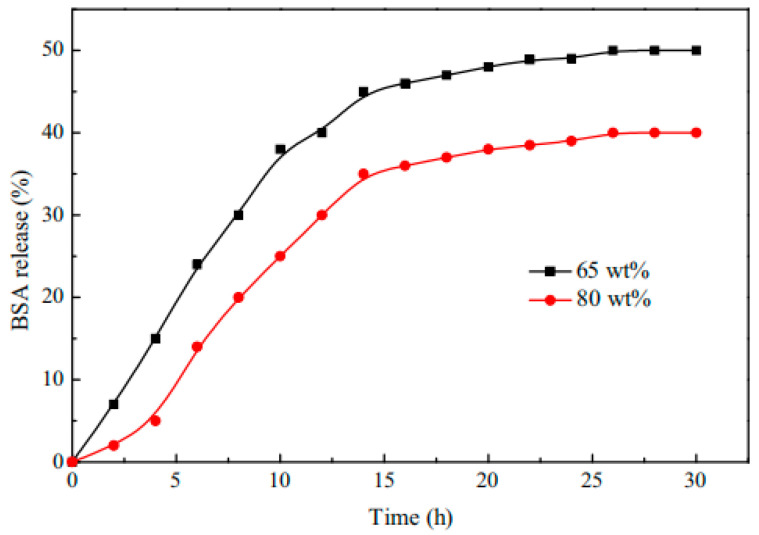
Cumulative release of BSA in gels formed by different concentration PMHD4 solutions “Reproduced with permission from Liu S., et al., Micellization and sol-gel transition of novel thermo- and pH-responsive ABC triblock copolymer synthesized by RAFT, J POLYM RES, 2018, 25(12)” [121].

**Figure 34 molecules-28-04246-f034:**
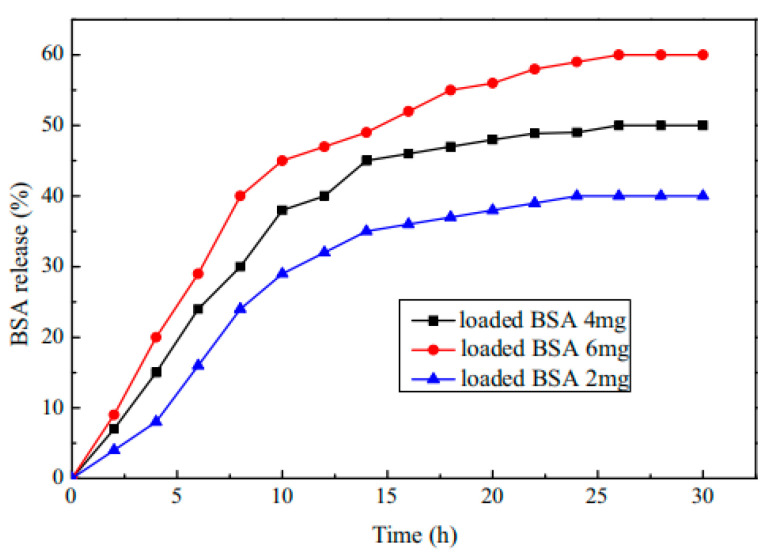
Cumulative release of BSA in PMHD4 hydrogel “Reproduced with permission from Liu S., et al., Micellization and sol-gel transition of novel thermo- and pH-responsive ABC triblock copolymer synthesized by RAFT, J POLYM RES, 2018, 25(12)” [121].

**Figure 35 molecules-28-04246-f035:**
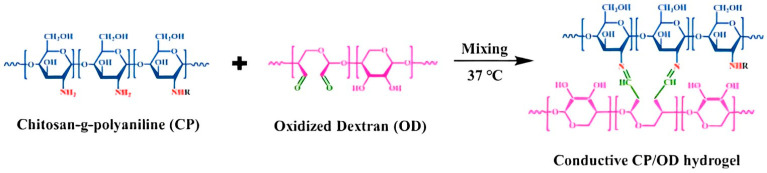
Synthesis of CP/OD hydrogels “Reproduced with permission from Qu J. et al., Injectable antibacterial conductive hydrogels with dual response to an electric field and pH for localized “intelligent” drug release, ACTA BIOMATER, 2018” [122].

**Figure 36 molecules-28-04246-f036:**
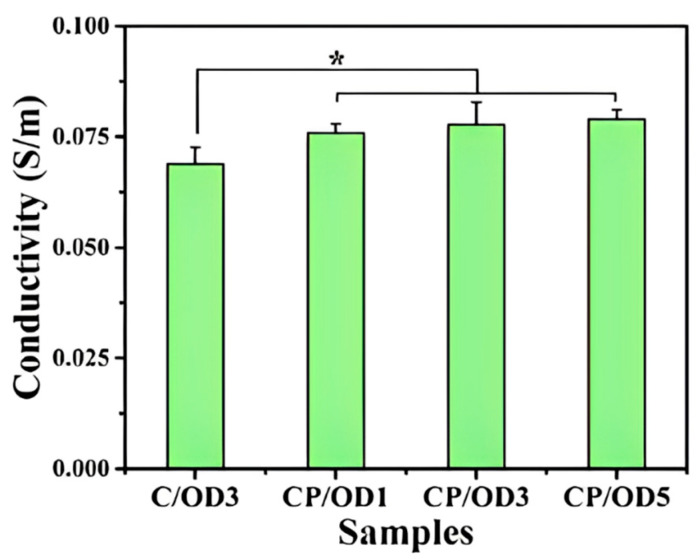
Conductivity of the CP/OD hydrogels, * = C/OD3 compares with CP/OD1, CP/OD3 and CP/OD5. “Reproduced with permission from Qu J. et al., Injectable antibacterial conductive hydrogels with dual response to an electric field and pH for localized “intelligent” drug release, ACTA BIOMATER, 2018” [122].

**Figure 37 molecules-28-04246-f037:**
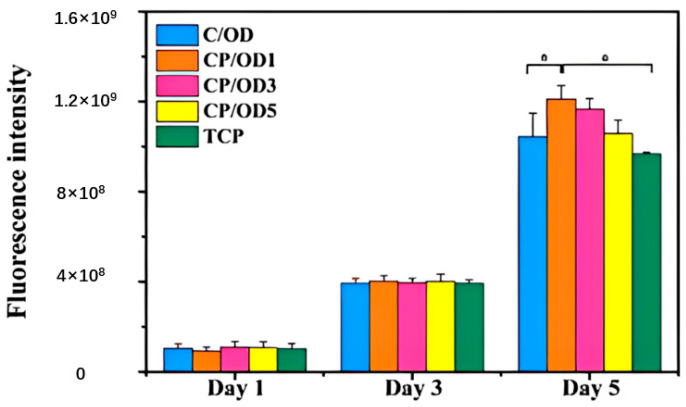
The proliferation of the L929 cells for hydrogels and TCP. The circles ° explained cell proliferation was higher in CP/OD1 hydrogels than in TCP and C/OD hydrogels). “Reproduced with permission from Qu J. et al., Injectable antibacterial conductive hydrogels with dual response to an electric field and pH for localized “intelligent” drug release, ACTA BIOMATER, 2018” [122].

**Figure 38 molecules-28-04246-f038:**
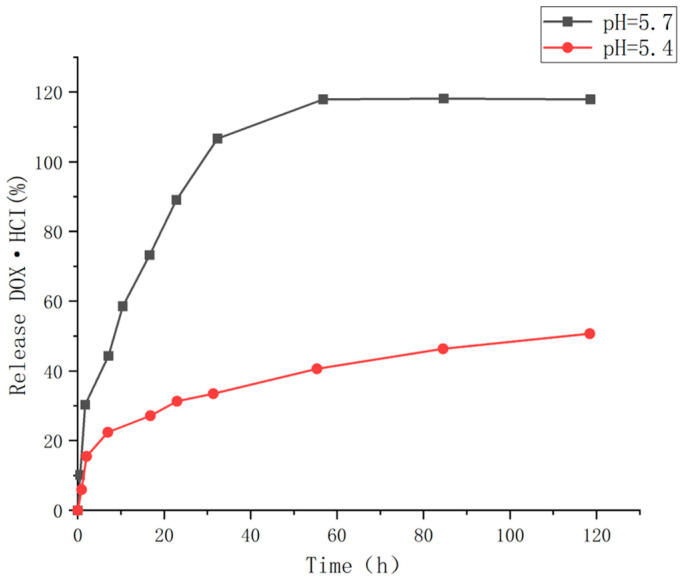
pH-dependent release of DOX from MPEG1K-PLYS2-PV7 hydrogel at 37 °C [124]. (adapted from preparation and properties of a temperature- and pH-responsive polypeptide hydrogel).

**Table 1 molecules-28-04246-t001:** Comparison of different types of intelligence polymer gels.

Type	Performance and Features	Application Field	Proposed Development Direction
Temperature response type	It responds quickly to changes in external ambient temperature and changes in physical or chemical properties.	Transport and release, biological organs and other fields.	Due to its poor mechanical strength and biocompatibility, it is necessary to improve its mechanical properties, diversification and controllable self-driven deformation while maintaining high response rate [48,49,50,51,52,53,54,55,56,57,58].
Light response type	It can respond quickly to chemical or physical changes under the action of light.	Industrial field and biomedical field, etc.	The research on photoresponsive gels is in its infancy, and its response mechanism needs to be further strengthened. Based on the understanding of intelligent light-responsive polymer gel materials, new light-sensitive materials can be synthesized by using the principles of polymer design and synthesis [59,60].
Electric field response type	Under the action of electric field stimulation, the gel will change in volume or shape (mainly the swelling, deswelling and bending deformation of the gel), so as to realize the transformation from electrical energy to mechanical energy.	Biological engineering, electronic materials and other fields.	Electric-field-driven polymer gels have a wide application prospect in the field of bioengineering, but the theory in this field is not mature, and the electric field-response of polymer gels is not well understood. It is necessary to establish accurate mathematical model on the basis of a large number of experimental data, and carry out a lot of research to promote the indepth development of basic theory and synthesis technology [77,78,79,80,81,82,83,84,85,86,87,88].
PH response type	Rapid response to changes in external pH, mutation.	Industrial field, biomedical field and so on.	Mechanical properties poor strength, unstable performance, poor biocompatibility, difficult to degrade. To study biodegradable gels, it is necessary to improve their mechanical properties while maintaining high response, and pay attention to the impact on the environment [89,90,91,92,93,94,95,96,97,98,99,100,101,102,103,104].
Chemical response type	The swelling behavior of chemical-influenced gels is mutated by the stimulation of specific chemicals, such as sugars.	Drug release, protein carrier, tissue engineering, biomedical fields, etc.	The mechanical properties are poor, the research is in the basic stage, the technology is not yet mature, and there are few kinds of chemical substances available. It is suggested to study these chemical substances more, prepare more functional intelligent polymer gels, and apply them in more fields [105,106,107,108,109,110,111,112,113,114,115].
Multiple response type	With a variety of single response performance synthesis, can simultaneously respond to a variety of external environmental stimuli; With double or even triple response characteristics.	Drug control release system, memory switch, artificial muscle, chemical memory, material separation, biomedical field, etc.	It is suggested that the sensitive mechanism should be further elucidated in future studies. The mechanical strength and response rate of gel can be improved through the coordination of different technologies. An environmentally sensitive polymer gel with biocompatibility and biodegradability is constructed from the perspective of bionics. Vigorously promote the industrialization and scale of gel production [116,117,118,119,120,121,122,123,124,125,126,127,128].

## Data Availability

Not applicable.

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
