# Peer review of "A Review of Research Progress on the Performance of Intelligent Polymer Gel"

_molecules, 2023, doi:10.3390/molecules28104246_

Round 1

Reviewer 1 Report

C1- Page 2, line 49: Here it should be briefly mentioned which defects occur and what kind of difficulties these defects cause in terms of applications of the intelligent materials.

C2-Page 2, line 84: what is the origin of the magneto-elastic properties, how they change or arise, what properties do the material acquire, giving them the potential to mimic muscle contraction.

C3-Page 5, line 132: “A series of biodegradable PECE triblock copolymers” what is PECE.

Page 5, line 141: Describe shortly the bottleneck problem.

Page 5, line 143: Check the reference work of 27 and redefine the clay XLG “Ning Luping[27] used NIPAM as a monomer, synthesized lithium magnesium silicate (XLG)”

Line 158: Please define what is MSN in PNIPA/MSN composite hydrogel.

C4-Page 7, line 197: Explain how a gel polymer binds to a photosensitive functional group, by what methods, what these groups are.

C5-Page 11, line 315: Specify why these properties of electric field sensitive gels arise. P( HEMA-co-AMPS ) polyelectrolyte described br Ref. Work 51 is sensitive to electric field due to which functional groups, please specify.

C6-Page 14, line 401: Why is cellulose-based materials mentioned first when mentioning pH-sensitive gels, why is the pH-sensitivity of cellulose interesting? It is stated that the use of biological materials has increased widely in recent years and the research of cellulose-based composites has become a hot direction.

C7-Page 15, line 427: Please give a brief explanation of what elastic modulus of smart gels should be in order for them to be used in injectable cell engineering scaffolds.

C8-Page 18, line 463: Explain the properties of gels that can be used in holographic processing technology. Revise Scheme 14 more clearly and in color and explain the pH response mechanism of the Hydrogel response in terms of functional groups.

C9-Page 20, line 56: What functional groups are important in the development of insulin self-adjusting gel delivery system with application prospects for diabetic patients as well as for glucose-responsive polymer gels.

Dear Editor

Molecules,

Thank you for your kind invitation to review the manuscript molecules-2362771 entitled “Performance research and development direction of intelligent polymer gel”.

The authors reviewed the studies from numerous and different perspectives based on the application of smart polymer gels, and mentioned in-depth research and applications and their important role in many fields.

I find the work original enough to be published in this journal, but still I have stated the criticisms that I have seen in the comments section.

Kind regards.

Author Response

Dear editor,

Thank you very much for your suggestion. The following is my revised content.The modified content has been highlighted in red in the text.I have replied to your modification suggestions in the attachment

Reviewer 2 Report

In this review manuscript, the authors present research progress on intelligent polymer gels and discuss the performance and future development directions of intelligent polymer gels. This review includes a comprehensive list of previous papers on the physical properties of stimuli-responsive gels. Therefore, the reviewers recommend that this manuscript be published in this journal as is.

Author Response

Thank you very much for your comments. 

The modified content has been highlighted in red in the text.

Reviewer 3 Report

The main concept of the paper seems to be interesting. However, Authors should pay attention to the following aspects:

1) The title of the paper should be changed to be more catchy.

2) First paragraph of the Introduction showing the definition of the intelligent polymer gel should be supplemented with adequate literature reference.

3) From the editorial viewpoint, there should be a pause between a word and a literature reference in brackets. Moreover, the literature reference in brackets should be before the punctuation mark, not after. 

4) Every abbreviation should be explained when it occurs for the first time in the text, e.g. NIPAM-HA-MA (line 53). It is additionally suggested to supplement the paper with additional subsection containing all abbreviations and their explanations.

5) There is too many spaces in line 67.

6) The notation of samples in Figure 1. are not clear.

7) All graphics in the paper should be named as "Figure", not as "Scheme".

8) From the editorial viewpoint, it is clearer to give the reference to the Figure (Figure shows...), next the Figure and finally the discussion concerning this Figure.

9) Line 373: it should be a superscript in the "cm2" unit.

10) Figure 11.: the font type and size should be in accordance with the requirements of the Journal. In general, all figures should be checked with this regard.

11) The size of all figures should be unified - e.g. Figs 25. and 26. differ from each other.

12) Table 1.: there are not any literature references. 

13) Section References should be prepared in line with the requirements of the Journal, e.g. the whole journal names should be replaced by their abbreviations. Additionally, it should be extended with more up-to-date references.

The paper should be re-checked linguistically and grammatically. There are some grammar mistakes. Additionally, Authors use sometimes very long sentences which make it difficult to fully understand the text and reduce its clarity.

Author Response

Thank you very much for your suggestion. The following is my revised content.The modified content has been highlighted in red in the text.I have replied to your modification suggestions in the attachment.

Round 2

Reviewer 1 Report

I reviewed the authors' responses to my questions. I think the revisions made are sufficient and appropriate.

Minor editing of English language required.